# Spirocyclic Motifs in Natural Products

**DOI:** 10.3390/molecules24224165

**Published:** 2019-11-17

**Authors:** Evgeny Chupakhin, Olga Babich, Alexander Prosekov, Lyudmila Asyakina, Mikhail Krasavin

**Affiliations:** 1Immanuel Kant Baltic Federal University, 236016 Kaliningrad, Alexandra Nevskogo 14, Russia; chupakhinevgen@gmail.com (E.C.); olich.43@mail.ru (O.B.); 2Kemerovo State University, 650000 Krasnaya, Kemerovo, Russia; rector@kemsu.ru (A.P.); alk_kem@mail.ru (L.A.); 3Saint Petersburg State University, 199034 Saint Petersburg, Russia

**Keywords:** natural products, spirocycles, chemical diversity, biological activity, privileged structures

## Abstract

Spirocyclic motifs are emerging privileged structures for drug discovery. They are also omnipresent in the natural products domain. However, until today, no attempt to analyze the structural diversity of various spirocyclic motifs occurring in natural products and their relative populations with unique compounds reported in the literature has been undertaken. This review aims to fill that void and analyze the diversity of structurally unique natural products containing spirocyclic moieties of various sizes.

## 1. Introduction

Natural products play the central role in drug discovery [1] due to their inherent biological activity and because have a wide span of structural diversity. Spirocyclic compounds have also occupied a special place in medicinal chemistry [2]. Spirocycles are thought to possess a good balance of conformational rigidity and flexibility to be, on one hand, free from absorption and permeability issues characteristic of conformationally more flexible, linear scaffolds. On the other hand, spirocycles are more conformationally flexible compared to, for example, flat aromatic heterocycles and can adapt to many proteins as biological targets; thus, increasing the chances of finding bioactive hits [3]. Spirocycles are distinctly three-dimensional and initial hits can be further optimized via manipulation of the molecular periphery whose three-dimensional positioning is well defined [4]. We thought it worthwhile to gain insight into the structural diversity of naturally-occurring spirocyclic compounds in relation to the information of their biological activity which would provide a new angle for designing novel bioactive, druglike compounds. Modern literature features a limited number of reviews devoted to total syntheses of spirocyclic natural products [5], including one for spirolactones [6] and one for spirooxyindoles [7]. Illustrative examples of approved natural-product drugs containing a spirocyclic motif include antifungal drug griseofulvin (**1**) and diuretic drug spironolactone (**2**). Interesting related compounds that have not achieved clinical approval include isochromanquinone antibiotic griseusin B (**3**) [8,9] and spirotriprostatin (**4**) [10] (Figure 1).

For the purpose of the analysis presented in this review, we considered the chemical diversity of structurally unique and well characterized (i.e., those whose structures were assigned using modern analytical techniques) spirocyclic compounds registered in the ChemBL or SciFinder databases, or the Dictionary of Natural Products (DNP). The occurrence of various ring combinations (A = any atom, mostly carbon or oxygen) selected for discussion in this review is presented in Table 1.

Considering the uneven distribution of the ring combination occurrence statistics (Table 1), the present review is structured according to the size of the [x.y.0] spirocyclic system. The review aims to cover either rare representatives of the spirocyclic systems that seldom occur in the natural product realm or only structurally-unique, representative compounds for those spirocyclic systems that are more widely populated with natural products reported in the literature, with an emphasis on their associated biological activities and the solid structure assignment techniques employed (structures assigned solely based on mass-spectrometric measurements are not taken into account).

## 2. [2.4.0] Spirocyclic System

Spirocyclic motifs containing a cyclopropane unit were found in some sesquiterpenes (**5**–**7**) which were isolated from the essential oils of South-American *Schinus terebinthifolius* fruit [11] (Figure 2). 

In 2017, a novel condensed [2.4.0] spirocycle (**8**) was reported [12]. It was isolated and characterized among the secondary metabolites of the *Helminthosporium velutinum* plant and was named cyclohelminthol X (Figure 3). This compound was shown to inhibit the growth of a human colon adenocarcinoma (COLO201) cell line with moderate potency (IC_50_ = 16 μM), and, much more potently (IC50 = 0.35 μM)—leukemia HL60 cell line [12].

Bioassay-guided separation of *Valerianae Radix* plant extract led to the isolation and characterization of valtrate (**9**), which inhibited Rev protein mediated transport of HIV-1 from the nucleus to cytoplasm (Figure 4). This compound also inhibited p-24 production of HIV-1 virus without any notable cytotoxicity displayed against MT-4 cells. The presence of the chemically labile oxirane ring as part of the generalized [2.4.0] spirocyclic system is likely critical for the observed inhibition, as **9** was shown to covalently interact with cysteine [13].

Additional two compounds (**31** and **32**) containing this and another ([4.4.0]) spirocyclic system are discussed in Section 7 of this review.

## 3. [2.5.0] Spirocyclic System

This group of spirocyclic natural products is represented by sesquiterpenoids illudins M and S (**10** and **11**, respectively) isolated from fungi, including the highly poisonous Jack-o′-lantern mushroom *Omphalotus illudens*. Compound **11** is currently in Phase II clinical trials against ovarian, prostate, and gastrointestinal cancers (Figure 5).

Structurally analogous to illudins are sesquiterpenes **12**–**14** isolated from fungus *Agrocybe aegerita* [14] also containing a [2.5.0] spirocyclic system (Figure 6). These compounds displayed antifungal activity against *Candida albicans* and *Candida kefyr*.

An oxirane-bearing sesquiterpene (−)-ovalicin (**15**) also containing a [2.5.0] spirocyclic system was isolated from fungus *Pseudorotium ovalis Stolk* [15]. It—and the structurally similar monoester fumagillin (**16**) displayed potent antiparasitic activities and are generally devoid of toxicity [16] (Figure 7). For both compounds **15** and **16**, total syntheses have been reported [17].

A [2.5.0] spirocyclic system is recognizable in the structure of duocarmycin SA (**17**) and duocarmycin A (**18**)—new antitumor antibiotics isolated from *streptomyces* sp. (Figure 8) [18].

## 4. [3.4.0] Spirocyclic System

This is an exceedingly rare type of spirocyclic motif encountered among natural products. The only compound reported in the literature to date containing such a spirocyclic system presented as a combination of a β-lactone and a pyrrolidine ring (**19**) was isolated from marine-derived *Streptomyces* strain collected in the southern area of the Korean Jeju Island [19] (Figure 9). This structurally intriguing compound displayed antibacterial activity.

## 5. [3.5.0] Spirocyclic System

The only spirocyclic combination of a four and six-membered rings represented in natural products is rather simple achiral 1-oxaspiro[3.5]nonan-7-ol substituted cleroindicin A (**20**) [20]. This compound was isolated from fungus *Clerodendrum japonicum* (Figure 10). 

## 6. [3.7.0] Spirocyclic System

This intriguing spirocyclic combination of four and eight-membered rings is represented in only four closely-related sesquiterpene bis-lactones, **21**–**24** (Figure 11), isolated from poisonous plants in the *Illicium* genus grown in China [21]. These structures could also be viewed as possessing a [3.5.0] spirocyclic motif.

## 7. [4.4.0] Spirocyclic System

Besides the approved diuretic spironolactone (**2**, vide supra), heteroatom-containing [4.4.0] spirocyclic motifs are widely represented by various lactones (Figure 12). 

The most structurally simple, naturally occurring spirocyclic lactone, 1,7-dioxaspiro[4.4.0]nonane or longianone (**25**) was isolated from higher fungi *Xylaria longiana* [22]. The absolute configuration of longianone was confirmed by stereoselective total synthesis [23]. Hyperolactones A (**26**) and C (**27**) isolated from *Hypericum chainens* plant [24] displayed antiviral activity [25]. The Nicolaou group reported a photochemical, [2 + 2]-cycloaddition based synthesis of a library based on natural product biyouyanagin (**28**) which allowed revising its originally reported absolute configuration [26]. (+)-Crassalactone D (**29**) is a styryl-lactone isolated from the leaves of *Polyalthia crassa* plant which displayed cytotoxic properties [27]. Pyrenolide D (**30**) is a highly oxygenated tricyclic spirolactone isolated from phytopathogenic fungus *Pyrenophora teres*, also displaying potent cytotoxicity [28]. Sesequiterpene levantenolide (**31**) also contained a [4.4.0] spirocyclic lactone moiety; it was isolated from tobacco grown in Turkey [29]. It exerted potent suppression of cytokine cascades and can, therefore, be considered a lead for anti-inflammatory drug development [30]. Complex polycyclic alkaloids represented by compound **32** were isolated from *Stemona* genus shrubs. These compounds contain a basic cyclopenta[1,2-b]pyrrolo[1,2-a]azepine scaffold and display promising anti-cough medicinal properties [31] (Figure 12). 

A [4.4.0] spirocyclic lactone moiety is found (in combination with a [2.4.0] spirocyclic oxirane) in limonoids **33**–**34**, which were recently isolated from *Trichilia connaroides* (Figure 13). For these compounds, some insights into a possible biosynthetic pathway have been provided. Likewise, these compounds were screened for various types of bioactivity and have been shown to inhibit NO production in a cellular model of inflammation (induced in RAW264.7 cell line with LPS) by 25.89% and 37.13% at 25 and 50 μM, respectively [32].

Studies of secondary metabolite structures in endophyte fungus *Penicillium purpurogenum* unveiled a series of unique sesquiterpene lactone compounds (**35**–**37**) containing spirocyclic combinations of three five-membered rings (Figure 14). All three compounds were screened against several cancer cell lines (melanoma M14, colon cancer HCT-116, glioma U87, ovary cancer A2780, stomach cancer BG-823, hepatoma Bel-7402, and lung cancer A549) and several pathogenic microorganisms (*Mycobacterium spegmatis* (ATCC70084), *Staphylococcus aureus* (ATCC25923), and *Staphylococcus epidermidis* (ATC12228)); however, no activity was detected at 50 μM [33].

Rather intriguing are the structures of curcumanolides **38**–**41**, natural [4.4.0] spirocyclic lactones recently isolated from *Curcuma heyneana*, a traditional medicinal plant of Indonesia (Figure 15) [34].

In the course of the thorough structural investigation of a series of iridoid glycosides isolated from the *Morinda citrifolia* plant, a revised structure was assigned. In particular, dehydromethoxygaertneroside (**42**), dehydroepoxymethoxygaertnoside (**43**), and citrifolinoside A (**44**) were shown to be structurally distinct compounds, all of which, however, possessed a [4.4.0] spirocyclic lactone moiety (Figure 16) [35].

Two diastereomers of spirophthalides, **45** and **46,** which possess a unique presentation of a [4.4.0] spirocyclic lactone, were recently isolated from a marine-sponge-derived fungus, *Setosphaeria* sp. (Figure 17) [36].

Unique spirocyclic dihydroindole-containing [4.4.0] spirocyclic lactones **47** and **48**, also possessing a quinazolone substituent, were identified among mycotoxins produced by *P. aethiopicum* (Figure 18) [37]. 

During a chemical and structural investigation of secondary metabolites of *Penicillium dangeardii*, a series of related [4.4.0] spirocyclic lactones (penicillactones A-C) **49**–**51** was identified. These possessed a complex molecular framework rich in carboxylate functionality and a well stereodefined substitution pattern around the spirocyclic core (Figure 19). Compounds **49**–**51** showed promise as leads for new antibiotic development. Additionally, penicillactones B and C (**50** and **52**, respectively) showed inhibition of the release of β-glucuronidase from polymorphonuclear leukocytes with ED_50_ values of 2.58 and 1.57 μM [38].

Rather intriguing and unique is the structure of spirocyclic hydantoins possessing a furanose unit. One of the first representatives of these natural products (hydantocidine **52**) was isolated from *Streptomyces hygroscopicus* (Figure 20). Hydantocidine displayed herbicidal properties which were linked to its ability to inhibit adenylate succinate synthase [39]. 

A unique presentation of a [4.4.0] spirocyclic system is featured in spirocyclic benzofuranones **53**–**55** isolated from ethanolic extracts of *Ganoderma Applanatum* (Figure 21) [40].

Not less interesting than the spirocyclic benzofuranones discussed above are natural products possessing a spirooxyindole motif. One of the first representatives of [4.4.0] spirocyclic compounds reported in the literature is spirotriprostatine (**56**), possessing moderate (IC_50_ = 197.5 μM) cytotoxic activity [41]. Naturally occurring spirooxyindoles were first isolated from plants *Apocynaceae* and *Rubiacae*, and from *Aspidosperma, Mitragyna, Ourouparia, Rauwolfia* and *Vinca* genera. These compounds can be further classified into two substructural classes: the tetracyclic secoyohimbane type (e.g., rhynchophylline (**57**)) and the pentacyclic heteroyohimbane type (e.g., formosanine (**58**)) (Figure 22) [42].

A very interesting class of natural products containing a [4.4.0] spirocyclic motif includes spiropseudoindoxyl alkaloids. Microbial transformation of the alkaloid mitragynine by the fungus *Helminthosporum* sp. was reported in 1974 to yield two major metabolites. The compounds were isolated from the biological milieu and their structures were elucidated as mitragynine pseudoindoxyl (**59**) and hydroxy mitragynine pseudoindoxyl (**60**) (Figure 23) [43]. These compounds were later shown to possess opioid analgesic activity by exerting mu agonism and delta antagonism while not recruiting β-arrestin-2 [44].

The [4.4.0] spirocyclic pseudoindoxyl motif represents a rather common feature in indole alkaloids, as can be illustrated by such examples as fluorocurine (**61**) [45], several diketopiperazines isolated from holothurianderived fungus *Aspergillus fumigatus* (**62a**–**d**) [46], brevianamide B (**63**) [47], and rauniticine pseudoindoxyl (**64**) [48] (Figure 24).

A structurally unique [4.4.0] spiroheterocyclic system is represented by a series of highly oxygenated lactone lactams (**65**–**69**) isolated from marine sediment-derived fungus *Aspergillus sydowi* D2–6 (Figure 25). Compounds **65**–**69** were shown to inhibit growth of adenocarcinoma cell line A549 with an IC_50_ value of 10 μM [49].

A wide diversity of heterocyclic spirocyclic scaffolds all belonging to the generalized [4.4.0] system (**70**–**73**) (Figure 26) have been isolated recently. Two regioisomeric phytoalexins—erucalexin (**70**) and its regioisomer (+)-1-methoxyspirobrassinin (**71**) were isolated from the wild crucifer *Erucastrum gallicum* [50].

Mycotoxins related to tryptoquialanine A (**71**) were isolated from *Penicillium* spp. and *Aspergillus clavatus* [51]. For tryptoquialanines, the biosynthetic pathway has been recently elucidated [25]. Another spirooxyindole lactone lactam compound **73** isolated from *Coix lachryma-jobi* L. has been recently reported and shown to possess activity against human lung cancer (A549) and colon carcinoma (HT-29 and COLO205) cell lines [52]. 

Secondary metabolite investigation of the liquid culture of entomogenous fungus *Isaria cateniannulata* led to the identification of a new spirocyclic compound **74** containing a 1,6-dioxaspiro[4.4]nonane moiety (Figure 27). The compound showed weak inhibitory activity against the HeLa cancer cell line [53].

Spirocyclic [4.4.0] tetrahydrofurans are featured in a series of twelve natural products **75a–l** dubbed bipolaricins (Figure 28). These compounds are ophiobolin-type tetracyclic sesterterpenes from a phytopathogenic *Bipolaris* sp. fungus. They were tested for HMGCoA reductase inhibition as well as anti-inflammatory and cytotoxic activities. The biological activity discovered provided the basis for considering these compounds as leads for antiinflammation and antihyperglycemic therapy developments [54].

An interesting type of [4.4.0] spirocyclic motif is present in fredericamycin A (**76**), an antitumor antibiotic produced by *Streptomyces griseus* (Figure 29) [55,56].

Summing up, the overall scaffold distribution within the general [4.4.0] spirocyclic system is shown in Figure 30.

Spirolactones are the most widely represented motifs in the [4.4.0] spirocyclic systems, with over 20 examples discussed above. Spirocyclic lactams are exemplified by 10 natural products. However, [4.4.0] spirocyclic lactam lactones and spirooxyindoles are much less common in the natural products and are represented by only a handful of examples. In terms of biological activity, the current data are mostly limited to cytostatic and antibacterial properties. The natural products isolated within the last 1–2 years are poorly investigated with regard to their biological properties.

## 8. [4.5.0] Spirocyclic System

Secondary metabolite investigation of *Teucrium viscidum* led to the identification of a [4.5.0] spirocyclic compound (**77**) possessing a unique skeleton [57]. A skeleton of similar complexity had only been featured once in the literature three decades before that [58] (Figure 31). 

The [4.5.0] spirocyclic motifs are featured in many natural terpenes. Recently, new spirocyclic triterpenoids **78**–**79** were isolated from *Leonurus japonicus* fruit (Figure 32). These compounds displayed moderately potent (IC_50_ < 10 μM) growth inhibition of five human cancer cell lines (stomach cancer BGC-823 and KE-97, hepatocarcinoma Huh-7, Jurkat T-cell limphoblasts, and breast adenocarcinoma MCF-7) [59].

Another example of an all-carbon [4.5.0] spirocyclic system is provided by spirocarolitone (**80**), recently isolated from *Ruptiliocarpon caracolito* [60] (Figure 33).

Structurally novel tricyclic-iridal triterpenoids belamcandanes A and B (**81** and **82**) (Figure 34) were recently isolated from *Belamcanda chinensis* and shown to possess moderate hepatoprotective properties. A possible biosynthetic pathway has been proposed [61]. 

New biologically active sesquiterpenoids **83**–**85** possessing an all-carbon [4.5.0] spirocyclic system were isolated from rhizomes of *Acorus calamus* (Figure 35). Compound **83** exhibited weak hepatoprotective activities against APAP-induced HepG2 cell damage [62].

The ethyl acetate soluble fraction of a MeOH extract of the dried stems and roots of *Capsicum annum* gave several new sesquiterpenoids, among which two [4.5.0] spirocyclic compounds termed canusesnols (**86**–**87**, Figure 36) were identified and evaluated for their cytotoxic activities [63].

Perhaps the most clinically advanced natural spirocyclic compound—spirocyclic benzofuran griseofulvin (**88**) isolated from *Penicillium griseofulvum* has been employed in clinical practice for therapy against ring worms [64] and was marketed by GlaxoSmithKline under the trade name Grisovin^TM^ [65] (Figure 37). 

Natural [4.5.0] spirocyclic lactones are characterized by a wide structural diversity and abundance of biological activities reported for them. These are exemplified by the mediator of mycoparasitism lambertollol C (**89**) [66], glycine-gated chloride channel receptor modulator (−)-ircinianin (**90**) [67], and terpenoid andirolactone (**91**) isolated from *Cedrus libanotica* [68] (Figure 38).

More examples of bioactive [4.5.0] spirocyclic lactones are provided by abyssomicins (**92a**–**c**, Figure 39), which were isolated from Actinobacteria and shown to inhibit *p*-aminobenzoate biosynthesis [69].

Antibacterial and antitumor compound lactonamycin Z (**93**) was isolated from *Streptomyces sanglieri* [70] and is an example of a [4.5.0] spirocyclic lactone embedded in a complex polycyclic system (Figure 40).

In 2015, Cech and co-workers reported new antibiotic spirocyclic lactone chaetocuprum (**94**) [71]. This compound was isolated from an endophyte fungus growing on the roots of wild *Anemopsis californica* plant which was traditionally used by North American tribes to treat infections and inflammation. Similarly, growing endophyte fungal parasites on the roots of *Chaetomium indicum* allowed Asai and Oshima [72] to isolate both epimers of spiroindicumide A and B (**95** and **96**, respectively) which feature an unprecedented spirocyclic lactone scaffold (Figure 41).

In addition to lambertollol C discussed above, two related epimeric compounds labertollol A (**97**) and B (**98**), also bearing a 4,8-dihydroxy-2,3,4-trihydronaphthalen-1-one scaffold and featuring a spirobutenolide moiety (Figure 42) were reported to possess high antifungal activity (IC_50_ = 0.5 μg/mL) [73].

Traditional Chinese medicinal plant *Rehmannia glutinosa* turned out to be a rich source of [4.5.0] spirocyclic lactones: massarigenin D (**99**), spiromassaritone (**100**), and paecilospirone (**101**) (Figure 43) which displayed potent (IC_50_ from 0.25 to 32 μg/mL) antifungal activity [74].

Perenniporide A (**102**) was the only spirocyclic lactone derivative of the naphthalenone family of natural products perenniporides A–D isolated from solid cultures of a fungus *Perenniporia* sp. inhabiting the larva of *Euops chinesis*, a phytophagous weevil with high host specificity to the medicinal plant *Fallopia japonica* (Figure 44) [75].

A [4.5.0] spirocyclic lactone moiety is featured in secochiliolide acid **103** (Figure 45), for which antiparasite activity was reported [76,77].

A rather unique [4.5.0] spirocyclic lactone moiety was identified in sesquiterpene abiespiroside A (**104**), which was isolated from Chinese tree *Abies dalavayi* (Figure 46). For this compound, anti-inflammatory activity was discovered [78].

A [4.5.0] spirocyclic lactone motif is featured in pathylactone A (**105**) isolated from marine sources, which demonstrated Ca^2+^ channel antagonistic activity (Figure 47) [79].

A whole series of spirolactones containing a terpenoid carane system (**106**–**110**) was reported as synthesized in enantioselective fashion (Figure 48). For these compounds, insect-feeding deterrent activity was reported [80].

In addition to the abundance of [4.5.0] spirocyclic lactones reported in the literature, some instances of spirocyclic tetrahydrofurans can be encountered. For example, 15-methoxycyclocalopin A (**111**) and isocyclocalopin A (**112**) were reported to be isolated from *Boletus calopus* [81]. Notably, compound **112** can be also considered a [5.5.0] spirocyclic hexahydropyran (Figure 49).

The structures of these compounds are reminiscent of spirocyclic dihydrofuran 8,9-dehydrotheaspirone, both enantiomers of which (**113a**–**b**) have been reported as volatile constituents of nectarines [82]. Their presence in the fruit was connected to some specific organoleptic properties of some kinds of nectarines (Figure 50)

Rather intriguing labdane-type diterpenoids (**114a**–**b**), epimeric to each other, isolated from the fruit of *Vitex agnus-castus* plant feature a unique skeleton consisting of both a [4.4.0] and a [4.5.0] spirocyclic tetrahydrofuran system (Figure 51) [83].

Rather unique is the structure of heliespirone **115** isolated from highly polar fractions of *Helianthus annuus* L. extract [84]. In this natural product, tetrahydrofuran forms a spirocyclic motif with a quinone-like moiety (Figure 52).

An oxygenated [4.5.0] spirocyclic framework is featured in several toxins, exemplified by arthropod toxin **116** (Figure 53) isolated from *Dinophysis acuta* and shown to potentiate erectile function [85].

Another example of similarly polyoxygenated [4.5.0] spirocyclic tetrahydrofuran is provided by quinochalcone **117**, named saffloquinoside A, isolated from *Carthamus tinctorius* (Figure 54) [86]. Compound **117** was evaluated in vitro for the inhibitory effect on the release of β-glucuronidase from rat polymorphonuclear neutrophils (PMNs) induced by the platelet-activating factor (PAF). It exhibited anti-inflammatory activity and the inhibitory rate was 54.3% (at 10^−5^ mol/L concentration).

Nitrogen-containing [4.5.0] spirocyclic systems are a lot more scarce compared to their oxygen-containing counterparts and can be exemplified by only two examples discussed below.

Alkaloid (±)-pandamarine (**118**) isolated as a major component from *Pandanus amaryllif olius* contains a [4.5.0] spirocyclic scaffold composed of a piperidine and a pyrollen-2-one rings (Figure 55) [87].

Another example of nitrogen-containing [4.5.0] spirocyclic system is provided by surugatoxin (**119**) isolated from the toxic Japanese ivory shell (*Babylonica japonica*) (Figure 56). This toxin suppresses the presynaptic nervous system [88]. Its total synthesis, in the racemic form, was achieved in 1994 by the Inoue group [89].

A [4.5.0] spirocyclic system is recognizable in spirostaphylotrichins which are spirocyclic γ-lactams mainly produced by several endophytic fungal strains of *Curvularia*, *Pyrenophora*, and *Staphylotrichum*. These are exemplified by spirostaphylotrichin X (**120**), characterized as an antiinfluenza agent targeting RNA polymerase PB2 [90], and spirostaphylotrichin W (**121**), investigated as a potential mycoherbicide for cheatgrass (*Bromus tectorum*) biocontrol [91] (Figure 57).

Summarizing this Section, the scaffold diversity stemming from the general [4.5.0] spirocyclic framework is comparable to that of the [4.4.0] spirocyclic system discussed earlier (Figure 30) and is shown in Figure 58.

## 9. [4.6.0] Spirocyclic System

As to the spirocyclic systems combining five and seven-memebred rings (the [4.6.0] spirocyclic system), spiro meroterpenoids spiroapplanatumines (**122**–**124**) isolated from the fruiting bodies of the fungus *Ganoderma applanatum* provide an eloquent example (Figure 59). Biological evaluation of the compounds disclosed that compound **124** inhibited JAK3 kinase with an IC_50_ value of 7.0 ± 3.2 μM [92].

In 2003, investigation of the neutral ether extracts of the fungus *Fomes cajanderi* led to the isolation of three novel ketal lactones named fomlactones A (**125**), B (**126**), and C (**127**) (Figure 60). The compounds clearly possess a [4.6.0] spirocyclic lactone moiety. However, their biological potential remains to be investigated [93].

A very unique spirocyclic [4.6.0] framework formed by a spiro[benzofuranonebenzazepine] skeleton is featured in natural products (±)-juglanaloid A (**128a**–**b**) and (±)-juglanaloid B (**129a**–**b**). These benzazepine alkaloids were isolated from the bark of *Juglans mandshurica*. Remarkably, both racemic natural products were successfully resolved by chiral separation and absolute configurations were unambiguously assigned (Figure 61). These enantiopure versions were screened for their in vitro inhibitory activities against self-induced Aβ_1-42_ aggregation using the Thioflavin T (Th-T) assay using curcumin as a reference compound. The compounds demonstrated promise acting as inhibitors of amyloid β aggregation [94].

Furthermore, in the last 1–2 years there has been an avalanche of new [4.6.0] spirocyclic structures reported in the literature. For examples, lanostane-type spirolactone triterpenoids **130a**–**c** isolated from *Ganoderma applanatum* (Figure 62) were reported to possess anti-hepatic fibrosis activities [95]. Interestingly, an additional [4.5.0] and [2.5.0] spirocyclic motif is recognizable in compounds **130b** and **130c**, respectively.

Another recent example (reported in 2019) of a [4.6.0] spirocyclic system is provided by grayanane diterpenoid auriculatol A (**131**) isolated from leaves of *Rhododendron auriculatum* (Figure 63). This compound is the first example of a 5,20-epoxygrayanane diterpenoid bearing a 7-oxabicyclo[4.2.1]nonane motif and a *trans/cis/cis/cis*-fused 5/5/7/6/5 pentacyclic ring system. Auriculatol A showed analgesic activity in the acetic acid-induced writhing test [96].

Finally another [4.6.0] spirocyclic lactone, seconoriridone A (isolated as a 7:1 epimeric mixture of (**132a**) and (**132b**)) was isolated in 2019 from *Belamcanda chinensis* (Figure 64). Although no biological activity was reported for this intriguing molecular structure, a plausible biosynthetic pathway was proposed [97].

The [4.6.0] spirocyclic system is amply exemplified in the natural products domain by the gelsenium alkaloids—gelsebanine (**133**), 14α-hydroxyelegansamine (**134**), 14α-hydroxygelsamydine (**135**) [98], 14-acetoxygelsenicine (**136**), 14-acetoxy-15-hydroxygelsenicine (**137**), 14-hydroxy-19-oxogelsenicine (**138**), and 14-acetoxygelseligine (**139**) [99] (Figure 65).

## 10. [4.7.0] Spirocyclic System

Spirocyclic natural products whose scaffolds contain rings larger than six-membered, e.g., [4.7.0] spirocyclic systems, are exceedingly rare. An eloquent example is provided by natural sugar-containing compounds phyllanthunin (**140**) recently isolated from an ethanol extract of the fruit of *Phyllanthus emblica* (Figure 66) [100]. 

Additionally, remarkably illustrative of the presence of [4.7.0] spirocyclic motifs in natural products, are portimines A (**141**) and B (**142**) isolated from the marine benthic dinoflagellate *Vulcanodinium rugosum* collected from Northland, New Zealand [101,102]. In addition to a [4.7.0] spirocyclic system, these compounds also contain a [4.5.0] spirocycle (Figure 67). Portimine has also been shown to induce apoptosis and reduce the growth of a variety of cancer cell lines at low nanomolar concentrations.

## 11. [5.5.0] Spirocyclic System

Among natural products containing a [5.5.0] spirocyclic motif, new spirocyclic chamigrane sesquiterpenes, merulinols B (**143**), C (**144**), E (**145**), and F (**146**) are notable examples (Figure 68). These compounds were isolated from basidiomycetous endophytic fungus XG8D associated with the mangrove *Xylocarpus granatum* [103]. The in vitro cytotoxicity of all compounds was evaluated against three human cancer cell lines, MCF-7, Hep-G2, and KATO-3. Compound **144** selectively displayed cytotoxicity against KATO-3 cells with an IC_50_ value of 35.0 μM.

Highly oxygenated acylphloroglucinol, hyperbeanol C (**147**), was isolated from the methanol extract of *Hypericum beanie* [104]. This compound contains an all-carbon [5.5.0] spirocyclic system, spiro[5.5.0]undec-2-ene-1,5-dione (Figure 69). The cytotoxicity of **147** against the cancer cell lines HL-60, SMMC-7721, PANC-1, MCF-7, K562, and SK-BR-3 was tested using the methyl thiazol tetrazalium (MTT) method with *cis*-platinum as the positive control. It exhibited modest cytotoxicity against K562 cells with an IC_50_ 16.9 μM.

Remarkable presentation of the (*R*)-1,7-dioxaspiro[5.5] undecane framework is found in nor-spiro-azaphilones, thielavialides A−D (**148**–**151**), and bis-spiro-azaphilone, thielavialide E (**152**) together with bis-spiro-azaphilone pestafolide A (**153**) (Figure 70). All these compounds were isolated from the endophytic fungal strain, *Thielavia* sp. PA0001, occurring in the healthy leaf tissue of aeroponically grown *Physalis alkekengi* [105].

A very similar [5.5.0] spirocyclic moiety can be found in the structure of pteridic acids C and F (**154** and **155**, respectively) isolated in 2017 from a culture broth of the marine-derived actinomycete *Streptomyces* sp. SCSGAA 0027 (Figure 71). While these compounds were seen as potential leads for antibacterial drug discovery, their extensive testing for antimicrobial activity against two gorgonian pathogenic fungal strains *Aspergullus versicolor* SCSGAF 0096 and *Aspergullus sydowii* SCSGAF 0035; a human pathogenic fungal strain *Candida albicans* SC5314; and two bacterial strains *Escherichia coli* and *Bacillus subtilis*, showed that the compounds had only a weak antimicrobial activity [106].

A unique [5.5.0] spirocyclic skeleton formed by a hexahydropyran and a pyrrolo[2,1-c]morpholine moieties is found in pollenopyrroside A (**156**) and B (**157**) isolated from bee-collected *Brassica campestris* pollen (Figure 72). The Chinese team who reported these natural products in 2010 also proposed a biosynthetic pathway that involves a reaction of 3-deoxy-d-fructose and 5-oxymethyl-2-formyl-pyrrole as the key step. Biological testing of these aldehyde compounds using the 3-(4,5-dimethylthiazol-2-yl)-2,5-diphenyltetrazolium bromide (MTT) method revealed that they possess no cytotoxicity against A549, Bel7420, BGC-823, HCT-8, and A2780 cancerous cell lines at 10 μg/mL [107].

Another unique [5.5.0] spirocyclic skeleton is noteworthy in the context of this review. Two structurally unique spirocyclic alkaloids **158** and **159** were isolated in 2007 from the halotolerant B-17 fungal strain of *Aspergillus variecolor* (Figure 73). Both compounds possessed an intriguing spirocyclic piperazin-2,5-dione moiety and exhibited cytotoxic properties [108].

Remarkably, in 2018, a very similar spirocyclic piperazin-2,5-dione variecolortin B (**160**) was isolated from the marine-derived fungus *Eurotium* sp. SCSIO F452 (Figure 74). The compound exhibited different antioxidative and cytotoxic activities. Interestingly, the same species gave rise to a compound possessing an even more seldomly-occurring spirocyclic moiety; namely, [5.6.0] (vide infra) [109].

The [5.5.0] spirocyclic system occurs very prominently in bioactive meroterpenoids **161a**–**e** and **162a**–**d** isolated in 2019 from mangrove-derived fungus *Penicillium* sp. (Figure 75). Several of these compounds showed growth inhibition activity against newly hatched larvae of *Helicoverpa armigera* Hubner with IC_50_ values ranging from 50 to 200 μg/mL, and some notable activity against *Caenorhabditis elegans* [110].

Workers of the ant *Carebarella bicolor* collected in Panama were found to contain the histrionicotoxin class of alkaloids with unusual 2,7-disubstituted-1-azaspiro[5.5]undecanol structures **163a**–**i** (Figure 76) [111].

## 12. [5.6.0] Spirocyclic System

An interesting group of natural products representative of this spirocyclic system are periplosides (**164**), a spiro-orthoester group-containing pregnane-type glycosides discovered in the course of phytochemical investigation of the root bark of *Periploca sepium* (Figure 77). The [5.6.0] spirocyclic orthoester core is distinctly modified with a steroid unit on one hand (R^1^) and with an oligosaccharide moiety on the other (R^2^). The compounds were evaluated for their inhibitory activities against the proliferation of T-lymphocytes. As a result, one specific compound (periploside C), the most abundant glycoside containing a spiro-orthoester moiety found in the plant, exhibited the most favorite selective index value (SI = 82.5). The inhibitory activity and the SI value appear to depend on the constitution of the saccharide chain [112].

The remarkable, from a structural perspective, spirolide G (**165**), was isolated from Danish strains of toxigenic dinoflagellate *Alexandrium ostenfeldii*. The toxicological profile of this compound was evaluated [113]. Interestingly, in addition to the spirocyclic [5.6.0] moiety in question, spirolide G (**165**) contains two others; namely, a [4.4.0] and a [4.5.0] motif (Figure 78).

Referring back to the chemical investigation of the marine-derived fungus *Eurotium* sp. SCSIO F452 discussed above in connection with compounds belonging to the [5.5.0] spirocyclic system, an intriguing [5.6.0] spirocyclic compound **166** (Figure 79) was also isolated from the same species [109]. This is a case of one species giving rise to a diversity of spirocyclic frameworks, underscoring the significance of spirocycles in the natural product realm. One particular example of such spirocycle diversity derived from a single organism is discussed in Section 13 below.

A [5.6.0] spirocyclic moiety is recognizable in the new sesquiterpene dimer vieloplain G (**167**) isolated in 2019 from the roots of *Xylopia vielana* (Figure 80). This compound showed considerable cytotoxicity against DU145 cells with an IC_50_ value of 9.5 μM [114].

## 13. [6.6.0] Spirocyclic System

This type of spirocyclic framework is exceedingly rare in the natural product domain, with only one example of unique 1-oxaspiro[6.6]tridecane **168**, a spirocyclic nortriterpenoid Spiroschincarin A isolated from the fruit of *Schisandra incarnate* (Figure 81) [115].

## 14. Plant Species Distinctly Rich in Diverse Spirocyclic Natural Products

Some cases when the same plant or microorganism gave rise to secondary metabolites with several structurally-diverse spirocyclic frameworks were discussed above. However, one recent example published in 2019, describing a chemical investigation of monoterpenoid indole alkaloids isolated from the roots of *Gelsemium elegans* (also briefly discussed in Section 8 of this review), stands out from the standpoint of hitherto unprecedented skeletal diversity [116]. In particular, the following spirocyclic frameworks were encountered among the natural products isolated from this species: [4.5.0]—featured in 19-oxogelsevirine (**169**), gelsevirine (**170**), and koumimine (**171**); [4.7.0]—featured in gelsedethenine (**172**); and a unique [4.8.0] system—featured in humantenine (**173**) and 19,20-epoxyhumantenine (**174**) (Figure 82). 

## 15. Conclusions and Perspectives

Spirocyclic scaffolds are omnipresent in the natural products domain. By analyzing the diversity of spirocyclic systems reported for natural products in the literature, one can appreciate an uneven distribution of such motifs according to the spirocycle type: certain motifs are more abundant than others and some are rather scarce, exemplified by only a handful of naturally occurring compounds. The most widespread are the [5.5.0], [4.5.0], and [4.4.0] spirocycles. In terms of associated bioactivities discovered and reported for spirocyclic products, these are mostly limited to the usual profiling in the context of antiproliferative, anti-inflammatory, and antimicrobial activities. However, the strong connections of spirocyclic frameworks to the natural product domain and their emerging privileged motif status in the synthetic drug discovery argues in favor of the need for more thorough panel profiling of all newly-discovered natural products, as novel and hitherto unprecedented bioactivity leads could be discovered. Certain scarcely-populated areas of the spirocyclic natural product space can be specifically developed into synthetic libraries and investigated for bioactivity. More spirocycles appear to have been discovered in the last 5–10 years, with an apparent advent of plant species giving rise to several types of spirocyclic frameworks in the course of their chemical investigation. The spirocyclic natural product discovery, therefore, appears to be on the rise and is likely to inspire new scaffolds for drug design and screening library development.

## Figures and Tables

**Figure 1 molecules-24-04165-f001:**
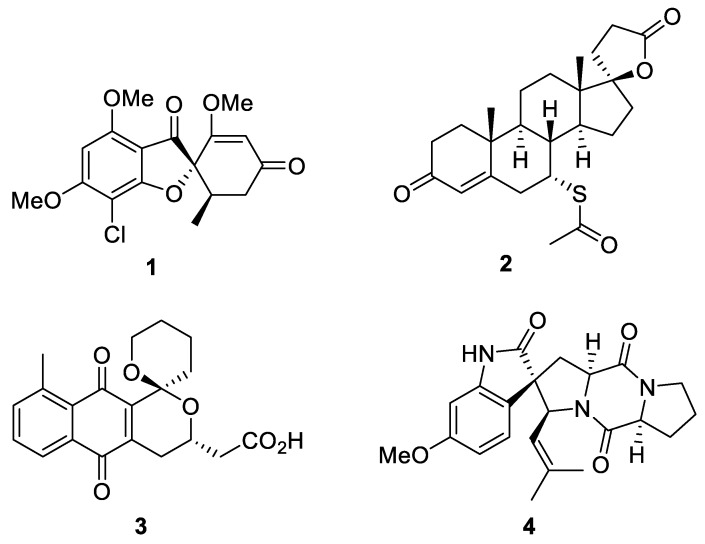
Examples of pharmaceutically important compounds bearing a spirocyclic motif.

**Figure 2 molecules-24-04165-f002:**
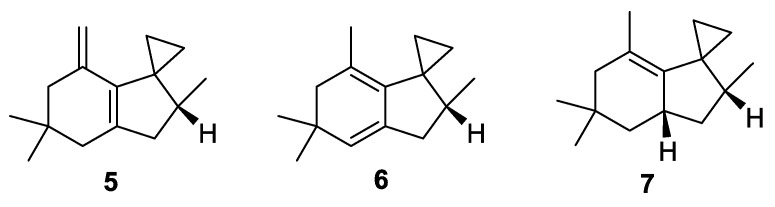
Sesquiterpenes from *Schinus terebinthifolius* fruit containing a [2.4.0] spirocyclic moiety.

**Figure 3 molecules-24-04165-f003:**
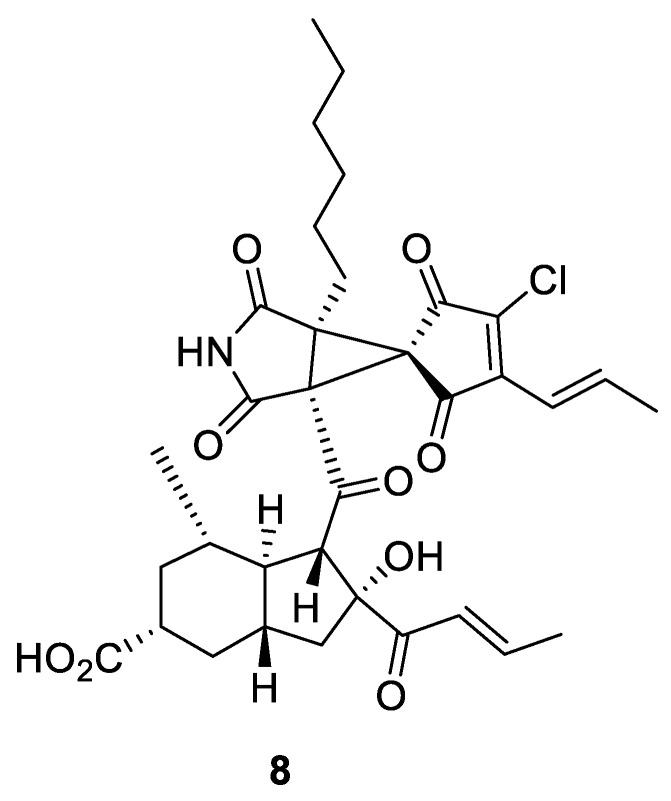
Cyclohelminthol X (**8**) from *Helminthosporium velutinum* plant.

**Figure 4 molecules-24-04165-f004:**
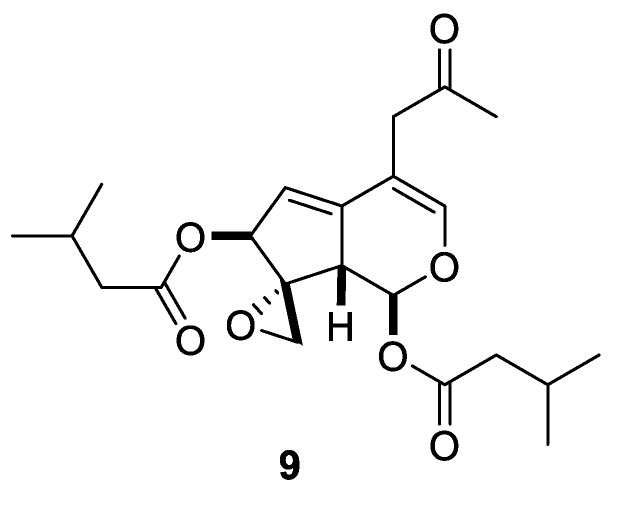
Valtrate (**9**) isolated from *Valerianae Radix* plant extract inhibiting HIV-1 transport.

**Figure 5 molecules-24-04165-f005:**
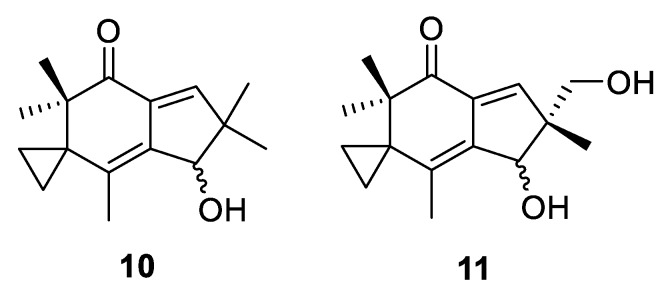
Structures of fungi-derived illudins M and S.

**Figure 6 molecules-24-04165-f006:**
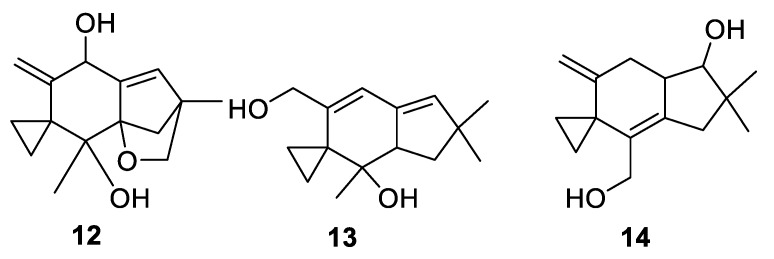
Structures of sesquiterpenes **12**–**14** isolated from fungus *Agrocybe aegerita*.

**Figure 7 molecules-24-04165-f007:**
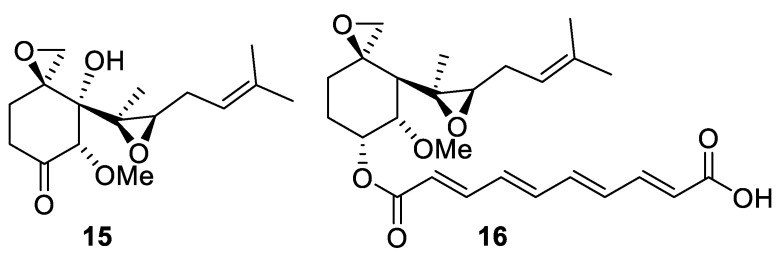
Structures of antiparasitic, fungus-derived (−)-ovalicin (**15**) and fumagillin (**16**).

**Figure 8 molecules-24-04165-f008:**
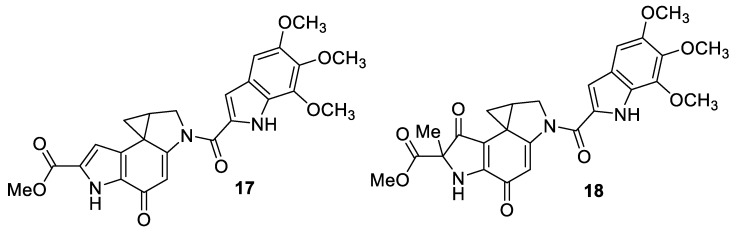
Structures of duocarmycin antitumor antibiotics.

**Figure 9 molecules-24-04165-f009:**
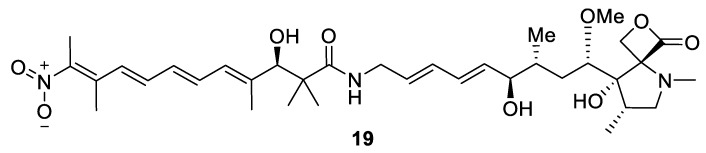
The only known natural product containing a [3.4.0] spirocyclic motif.

**Figure 10 molecules-24-04165-f010:**
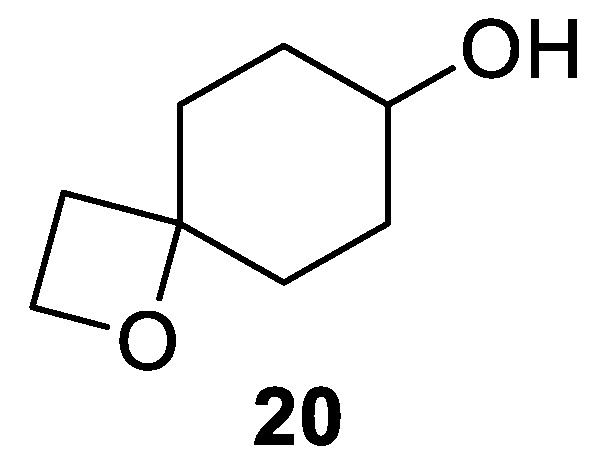
Cleroindicin A isolated from fungus *Clerodendrum japonicum*.

**Figure 11 molecules-24-04165-f011:**
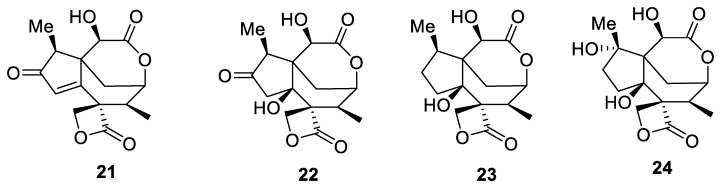
Sesquiterpene bis-lactones isolated from *Illicium* plants containing a [3.7.0] spirocyclic motif.

**Figure 12 molecules-24-04165-f012:**
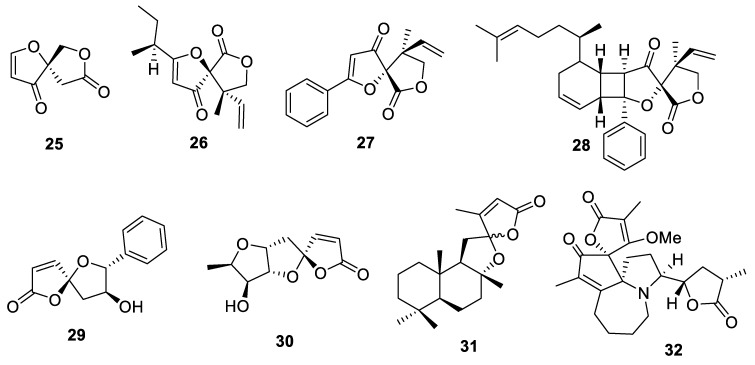
Various naturally occurring [4.4.0] spirocyclic lactones.

**Figure 13 molecules-24-04165-f013:**
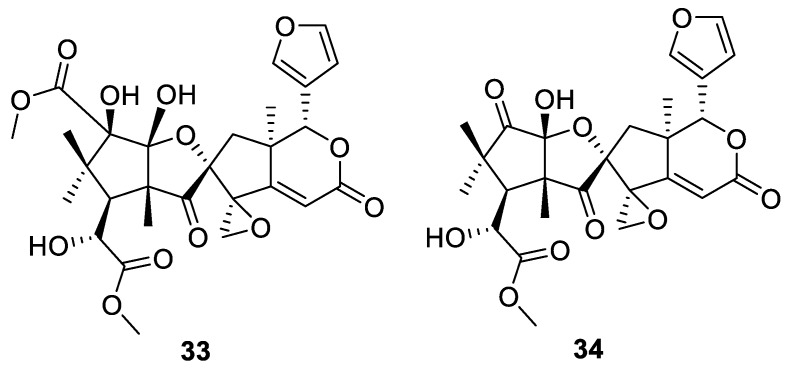
Limonoids **31**–**32** containing both a [4.4.0] and a [2.4.0] spirocyclic system.

**Figure 14 molecules-24-04165-f014:**
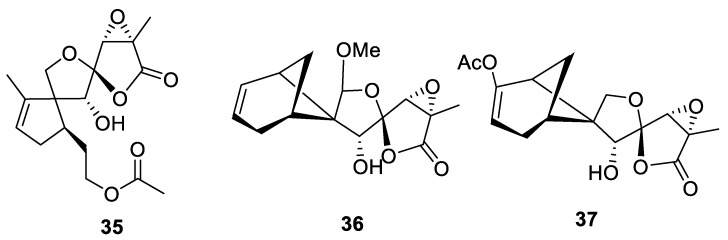
Tricyclic spirolactones (incorporating two [4.4.0] spirocyclic systems) isolated from *Penicillium purpurogenum*.

**Figure 15 molecules-24-04165-f015:**
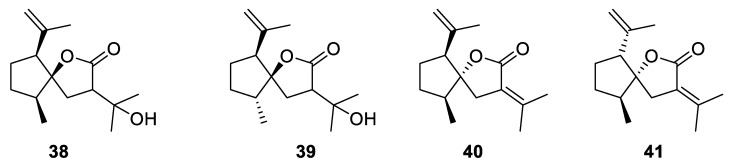
New spirocyclic curcumanolides possessing a [4.4.0] spirocyclic system each, isolated from *Curcuma heyneana*.

**Figure 16 molecules-24-04165-f016:**
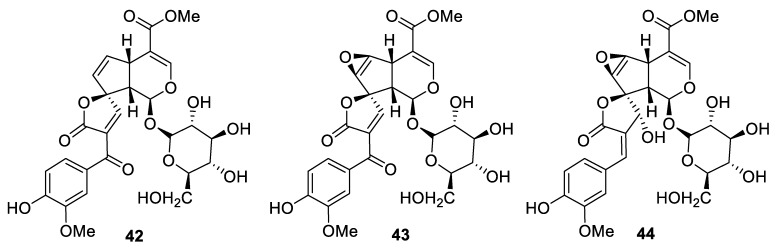
New iridoid glycosides isolated from *Morinda citrifolia* plant.

**Figure 17 molecules-24-04165-f017:**
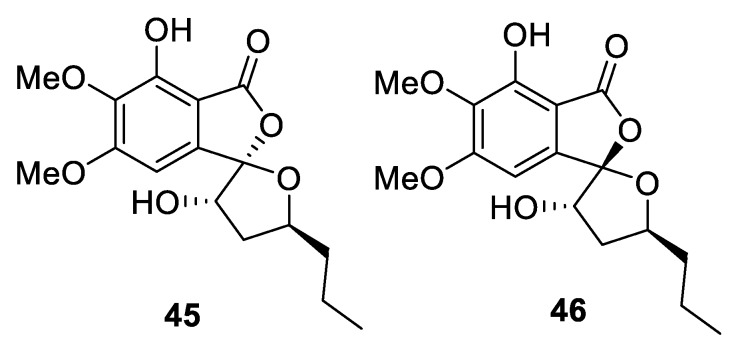
Diastereomeric spirophthalides recently isolated from marine-sponge-derived fungus *Setosphaeria* sp.

**Figure 18 molecules-24-04165-f018:**
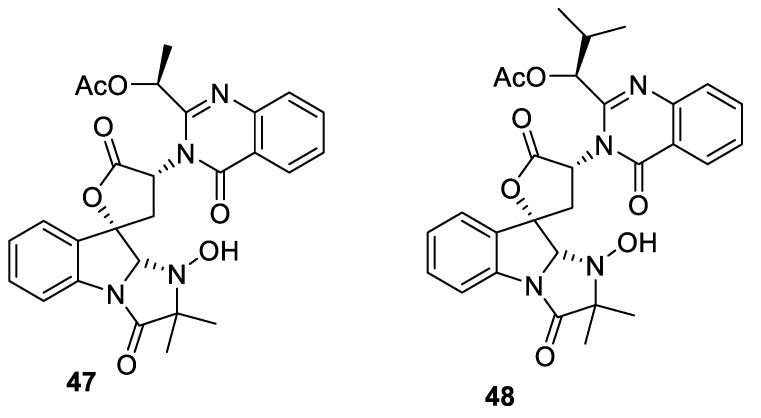
Spirocyclic mycotoxins produced by *P. aethiopicum*.

**Figure 19 molecules-24-04165-f019:**
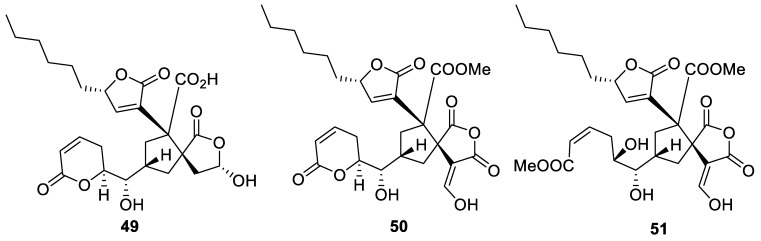
Spirocyclic lcatones isolated from *Penicillium dangeardii*.

**Figure 20 molecules-24-04165-f020:**
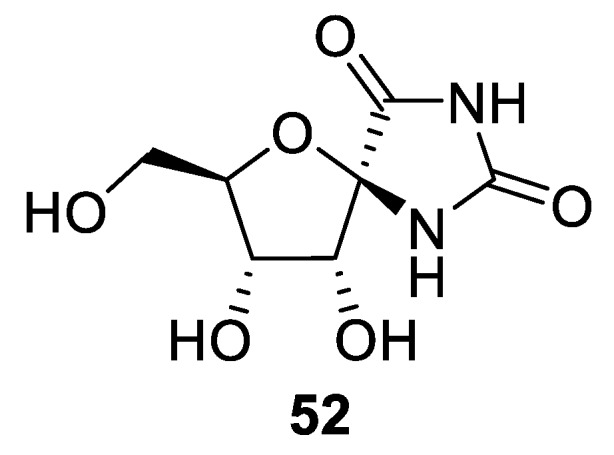
Structure of herbicidal hydantocidine isolated from *Streptomyces hygroscopicus*.

**Figure 21 molecules-24-04165-f021:**
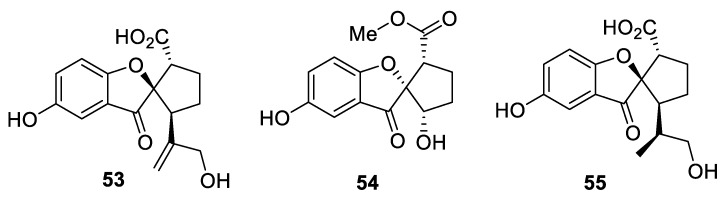
Spirocyclic benzofuranones isolated from *Ganoderma Applanatum*.

**Figure 22 molecules-24-04165-f022:**
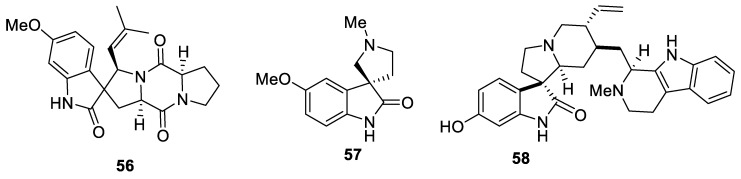
Examples of naturally occurring spirooxyindoles.

**Figure 23 molecules-24-04165-f023:**
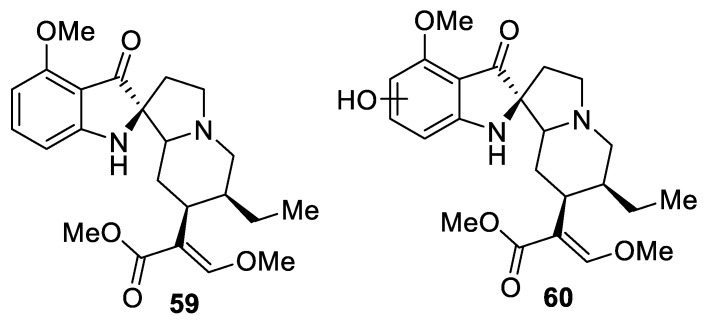
Structures of mitragynine pseudoindoxyl (**59**) and hydroxy mitragynine pseudoindoxyl (**60**).

**Figure 24 molecules-24-04165-f024:**
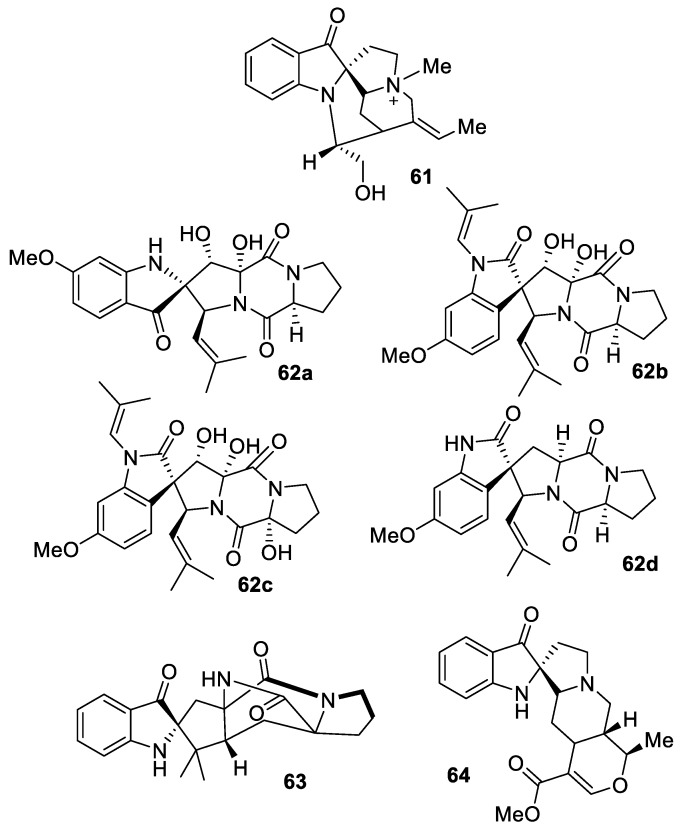
Structures of [4.4.0] spirocyclic pseudoindoxyl alkaloids fluorocurine (**61**), fungus-derived diketopiperazines (**62a**–**d**), brevianamide B (**63**), and rauniticine pseudoindoxyl (**64**).

**Figure 25 molecules-24-04165-f025:**
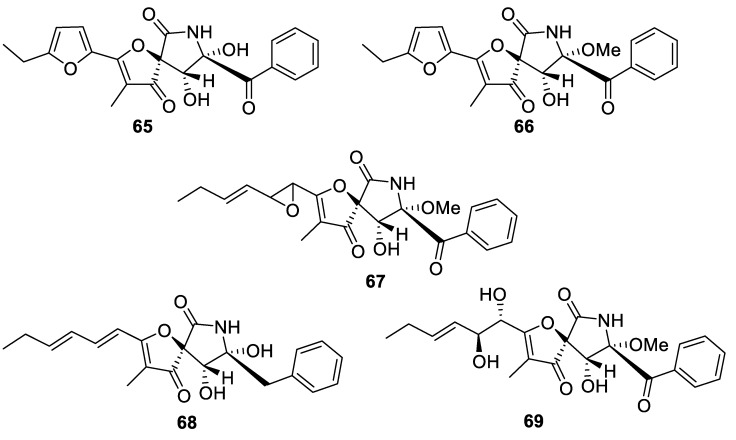
Members of a family of spirocyclic lactone lactams isolated from marine sediment-derived fungus *Aspergillus sydowi* D2–6.

**Figure 26 molecules-24-04165-f026:**
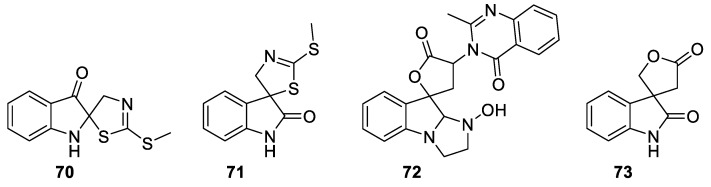
Natural products illustrating the range of heterospirocyclic [4.4.0]-sized motifs.

**Figure 27 molecules-24-04165-f027:**
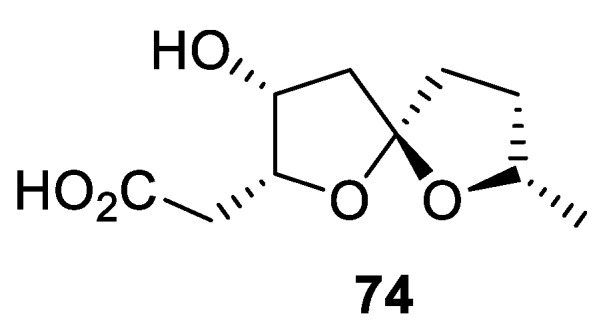
1,6-Dioxaspiro[4.4]nonane secondary metabolite isolated from entomogenous fungus *Isaria cateniannulata*.

**Figure 28 molecules-24-04165-f028:**
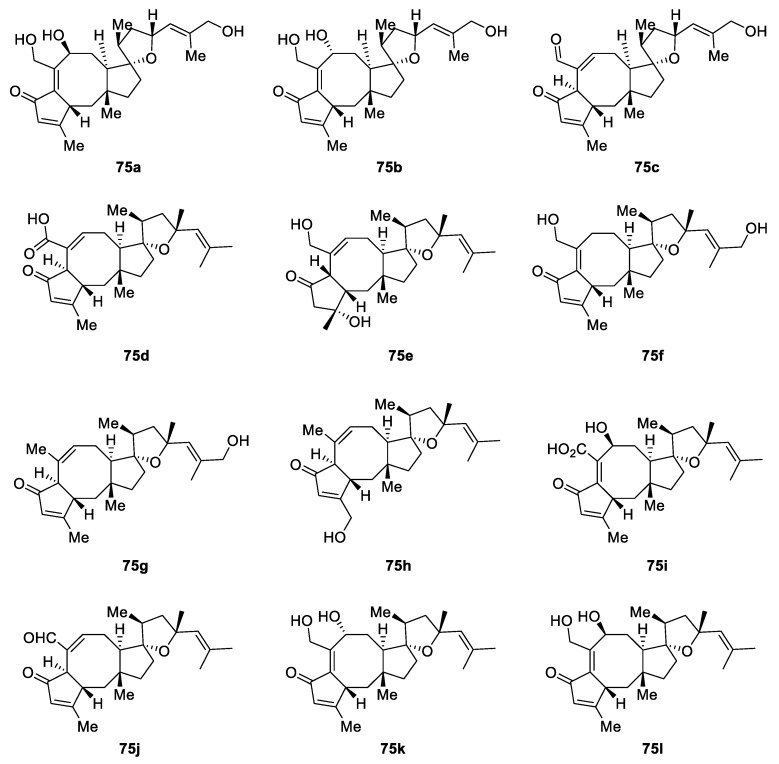
Bipolaricins from phytopathogenic *Bipolaris* sp. fungus.

**Figure 29 molecules-24-04165-f029:**
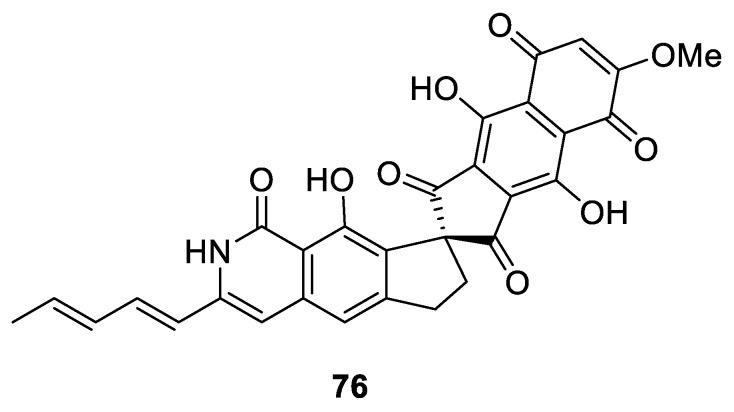
Structure of fredericamycin A possessing a [4.4.0] spirocyclic motif.

**Figure 30 molecules-24-04165-f030:**
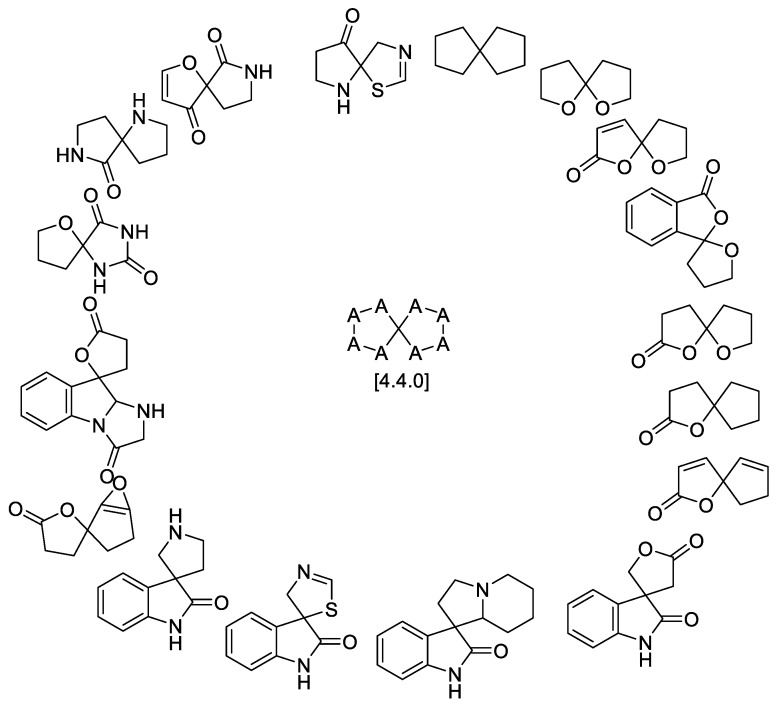
Overall diversity of [4.4.0] spirocyclic scaffolds represented in the natural products domain.

**Figure 31 molecules-24-04165-f031:**
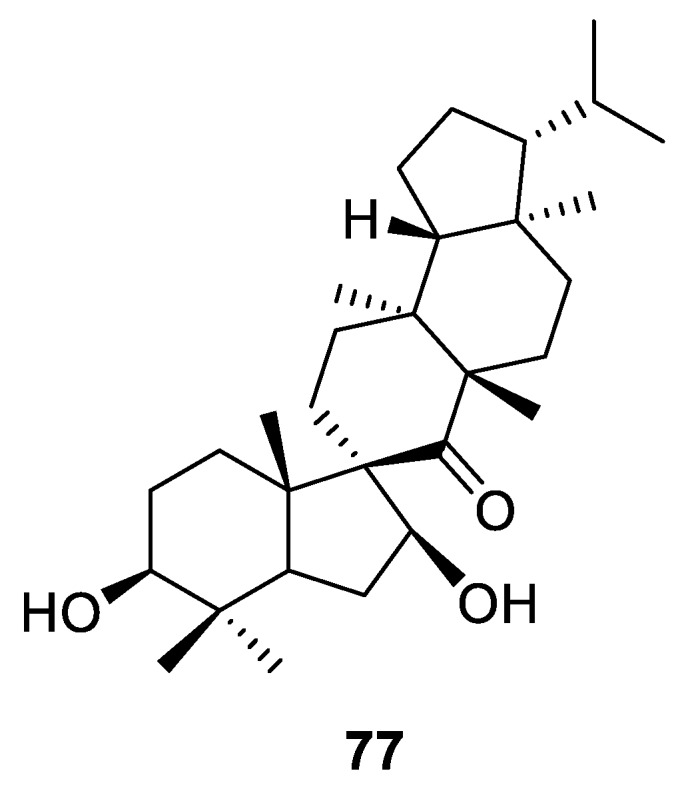
Spiro pentacyclic secondary metabolite isolated from *Teucrium viscidum*.

**Figure 32 molecules-24-04165-f032:**
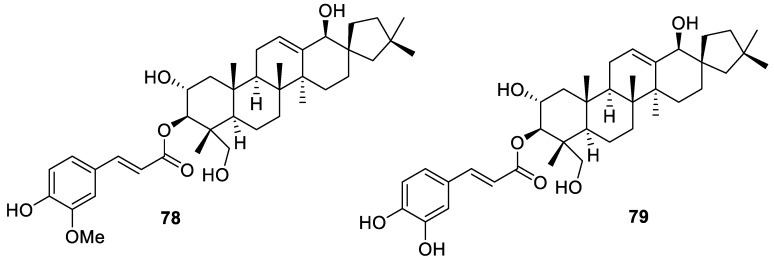
New spirocyclic triterpenoids isolated from *Leonurus japonicas*.

**Figure 33 molecules-24-04165-f033:**
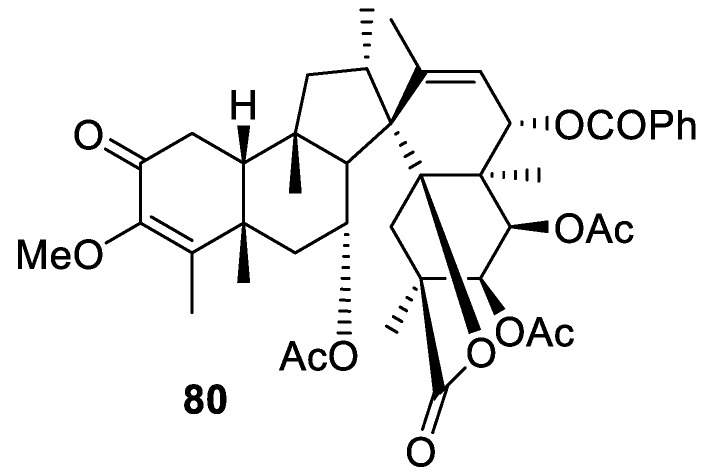
Spirocarolitone isolated from *Ruptiliocarpon caracolito*.

**Figure 34 molecules-24-04165-f034:**
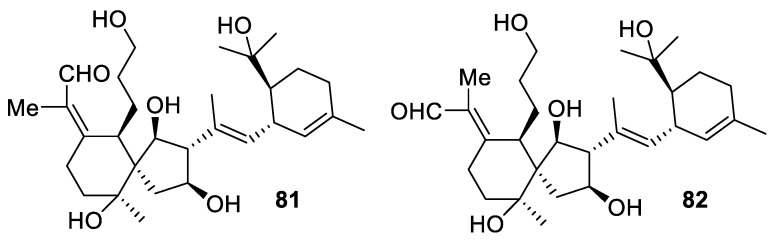
Structurally novel triterpenoids isolated from *Belamcanda chinensis*.

**Figure 35 molecules-24-04165-f035:**
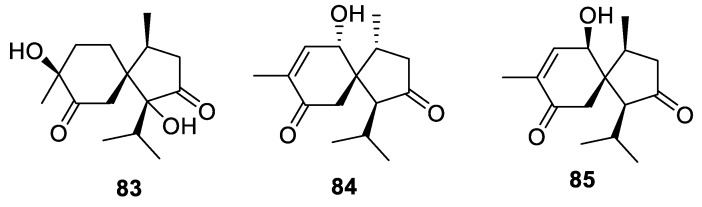
The [4.5.0] spirocyclic sesquiterpenoids from rhizomes of *Acorus calamus*.

**Figure 36 molecules-24-04165-f036:**
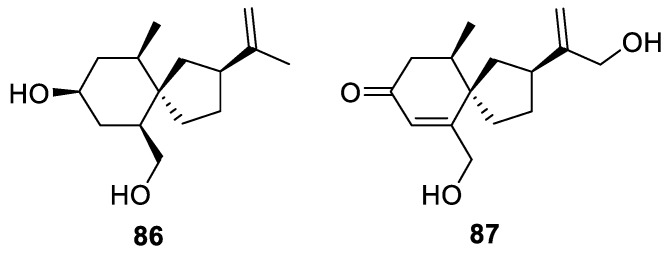
Canusesnols from *Capsicum annum*.

**Figure 37 molecules-24-04165-f037:**
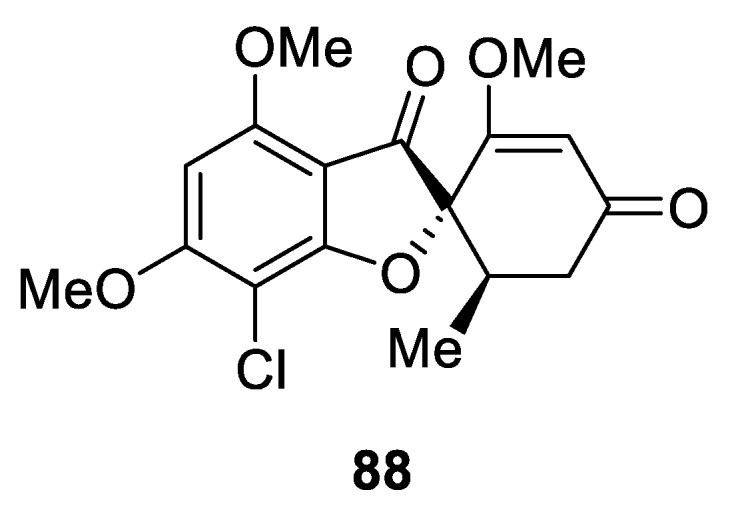
Anti-ring worm drug griseofulvin (Grisovin^TM^).

**Figure 38 molecules-24-04165-f038:**
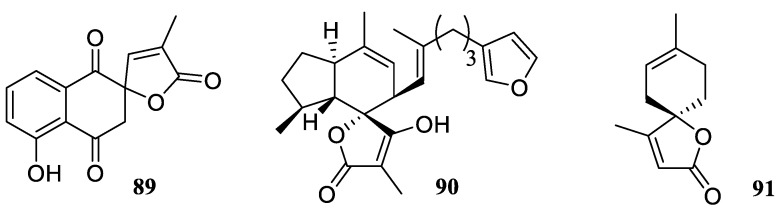
Examples of natural [4.5.0] spirocyclic lactones.

**Figure 39 molecules-24-04165-f039:**
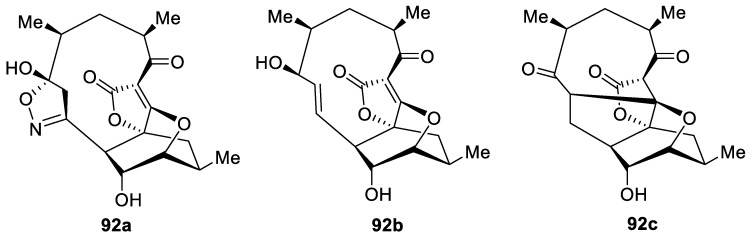
Abyssomycins from Actinobacteria.

**Figure 40 molecules-24-04165-f040:**
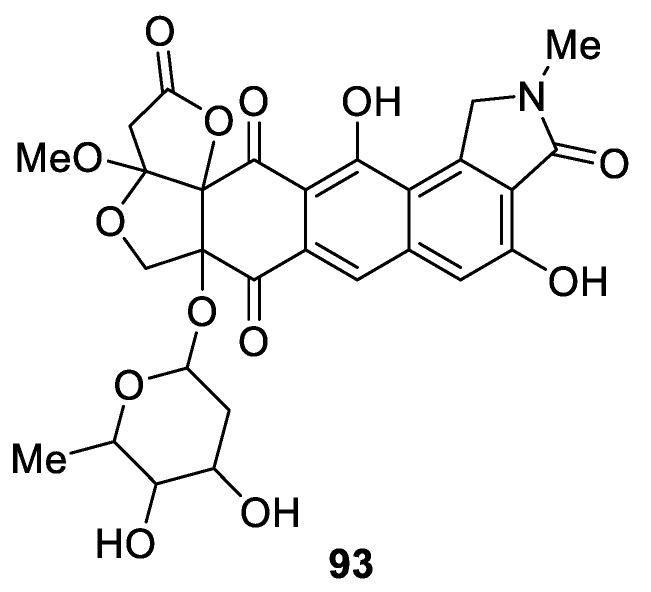
Antibacterial and antitumor compound lactonamycin Z isolated from *Streptomyces sanglieri*.

**Figure 41 molecules-24-04165-f041:**
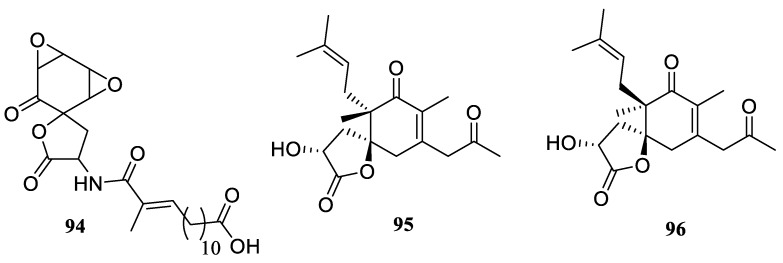
Spirocyclic lactones isolated from endophyte fungal parasites.

**Figure 42 molecules-24-04165-f042:**
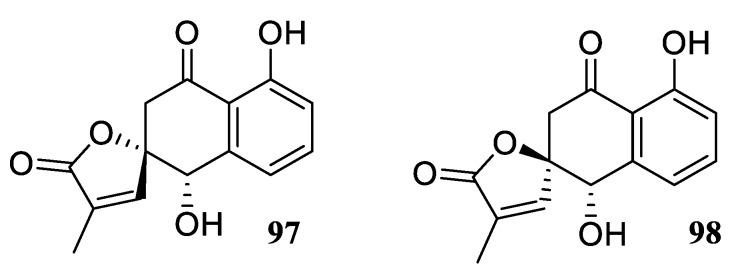
Lambertollols A and B.

**Figure 43 molecules-24-04165-f043:**
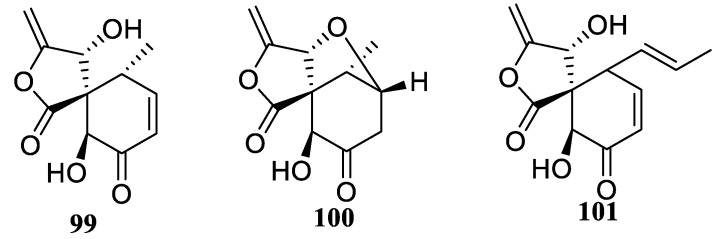
Spirocyclic lactones from traditional Chinese medicinal plant *Rehmannia glutinosa*.

**Figure 44 molecules-24-04165-f044:**
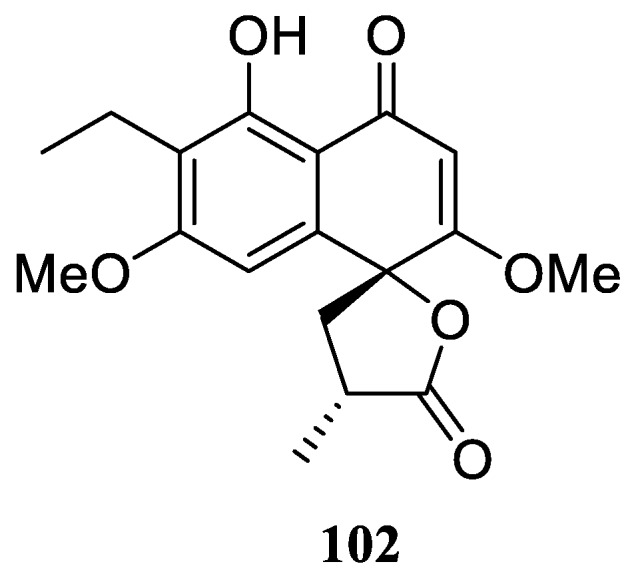
Perenniporide A, the only spirocyclic lactone of the perenniporide family.

**Figure 45 molecules-24-04165-f045:**
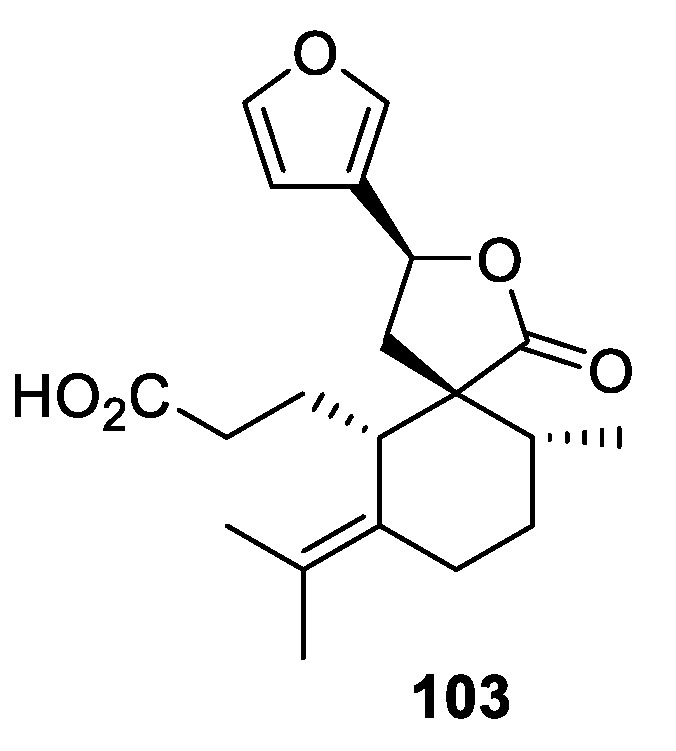
Secochiliolide acid.

**Figure 46 molecules-24-04165-f046:**
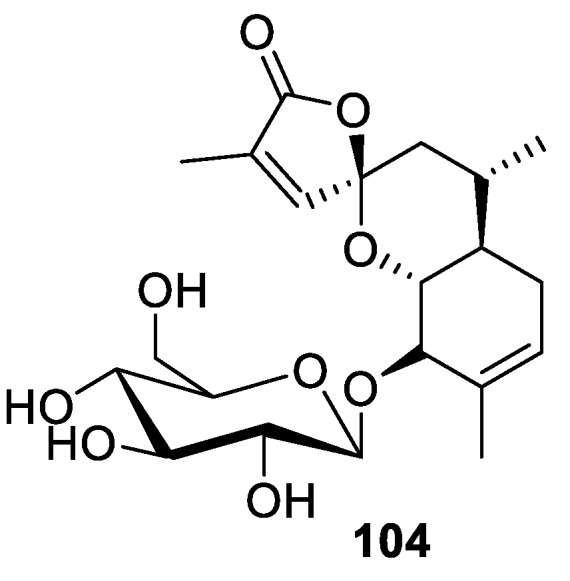
Abiespiroside A isolated from Chinese tree *Abies dalavayi*.

**Figure 47 molecules-24-04165-f047:**
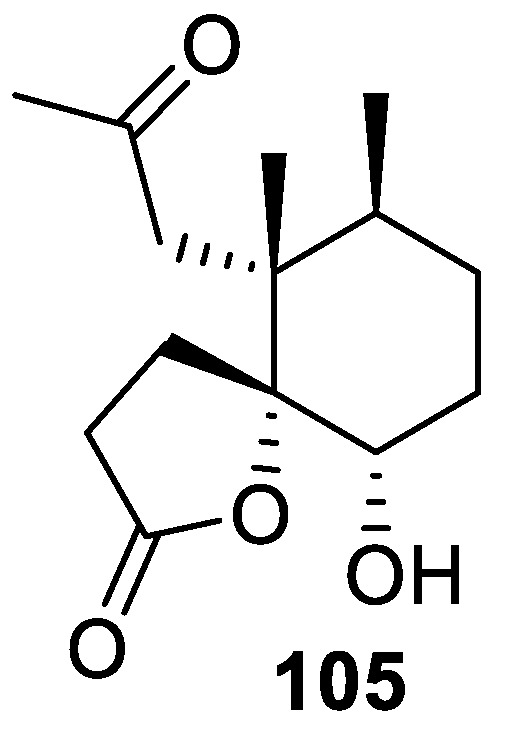
Pathylactone A isolated from marine sources.

**Figure 48 molecules-24-04165-f048:**
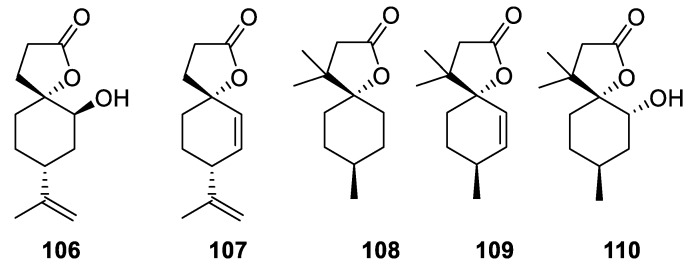
Spirocyclic carane lactones with insect-feeding deterrent activity.

**Figure 49 molecules-24-04165-f049:**
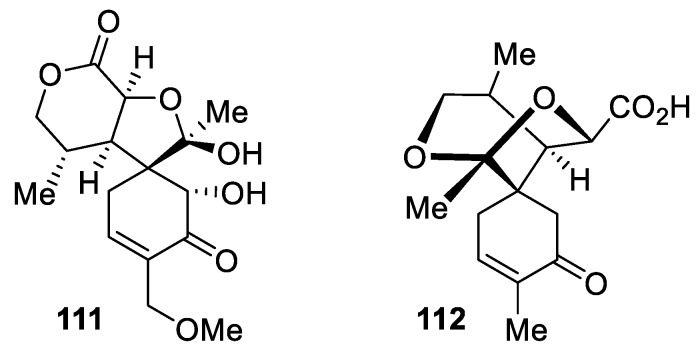
Natural spirocyclic tetrahydrofurans.

**Figure 50 molecules-24-04165-f050:**
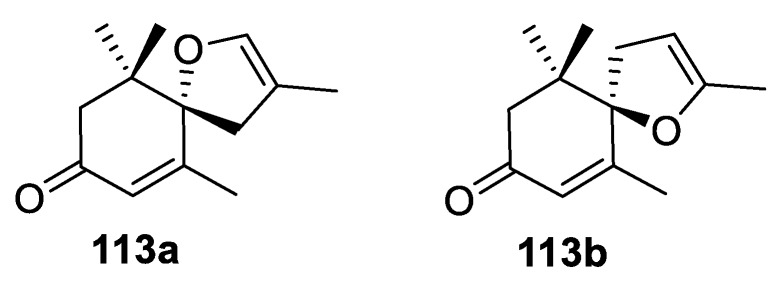
Enantiomers of [4.5.0] spirocyclic dihydrofuran 8,9-dehydrotheaspirone reported in the literature.

**Figure 51 molecules-24-04165-f051:**
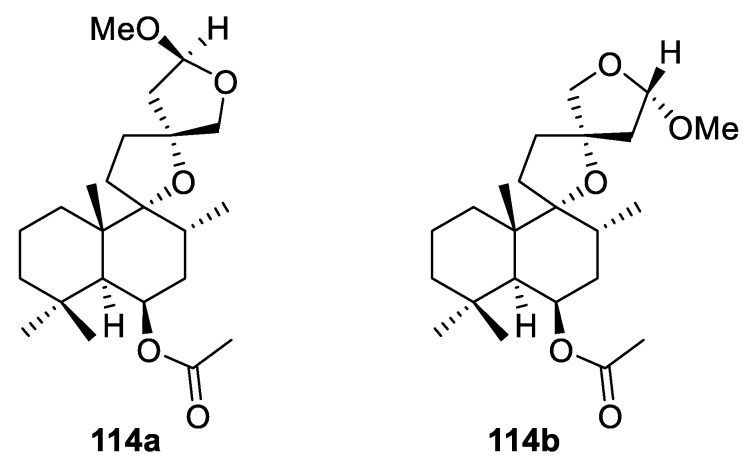
Spirocyclic labdane–type diterpenoids isolated from the fruit of *Vitex agnus-castus*.

**Figure 52 molecules-24-04165-f052:**
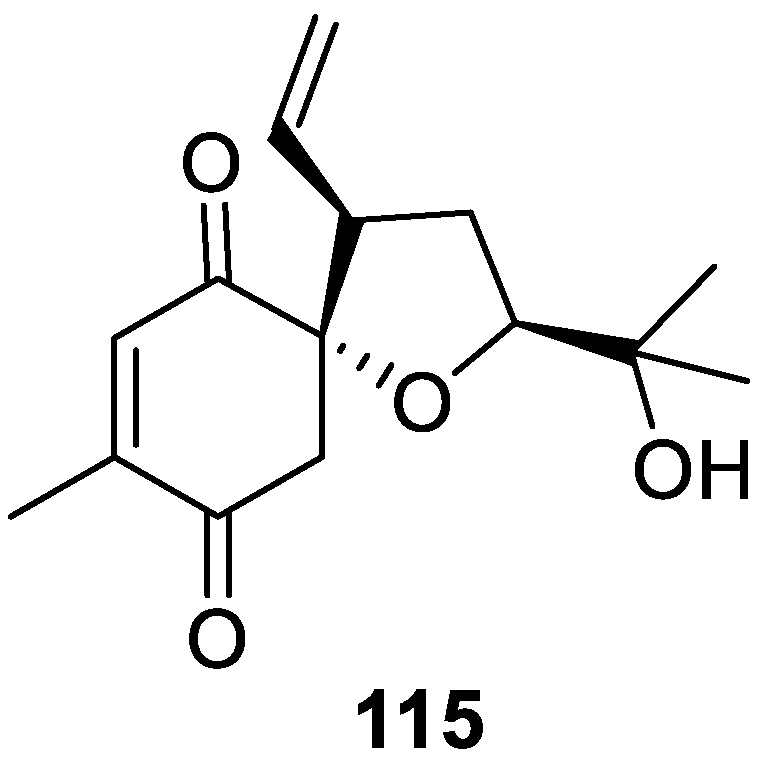
Spirocyclic natural product heliespirone featuring a tetrahydrofurane and a quinone-like moiety.

**Figure 53 molecules-24-04165-f053:**
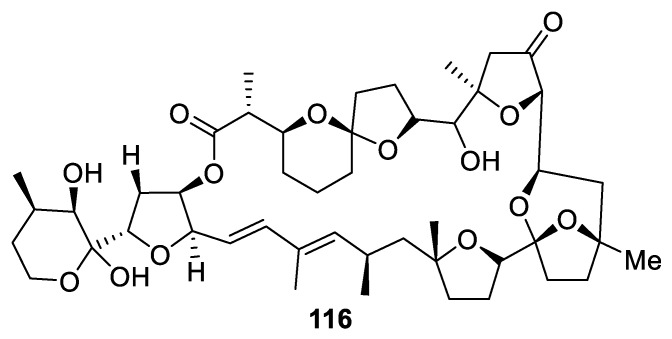
Erectile function-potentiating toxin featuring a [4.5.0] spirocyclic motif.

**Figure 54 molecules-24-04165-f054:**
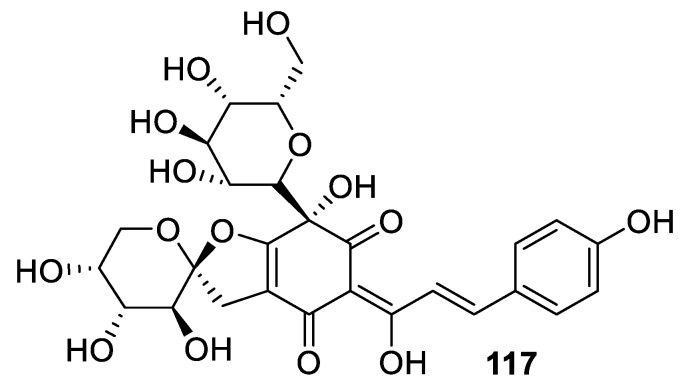
[4.5.0] spirocyclic quinochalcone saffloquinoside A isolated from *Carthamus tinctorius*.

**Figure 55 molecules-24-04165-f055:**
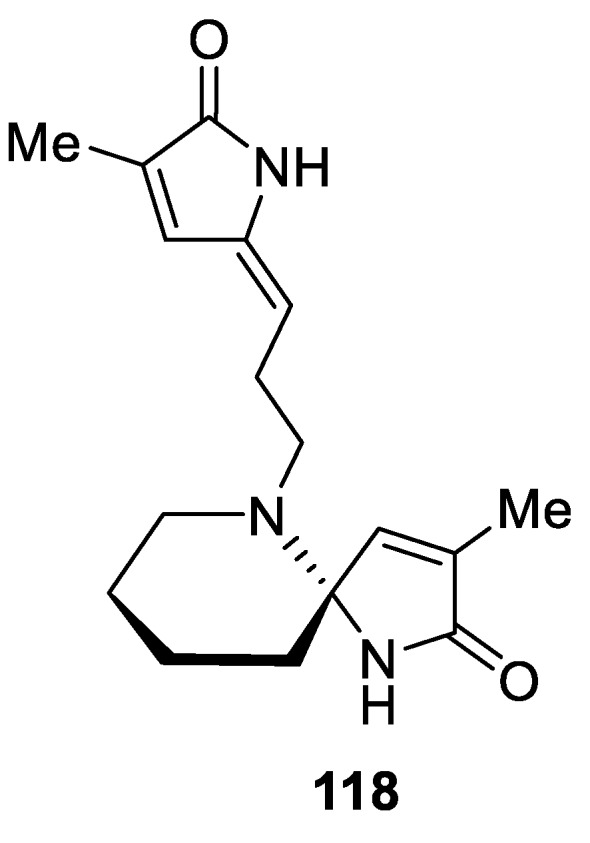
Alkaloid (±)-pandamarine isolated from *Pandanus amaryllif olius*.

**Figure 56 molecules-24-04165-f056:**
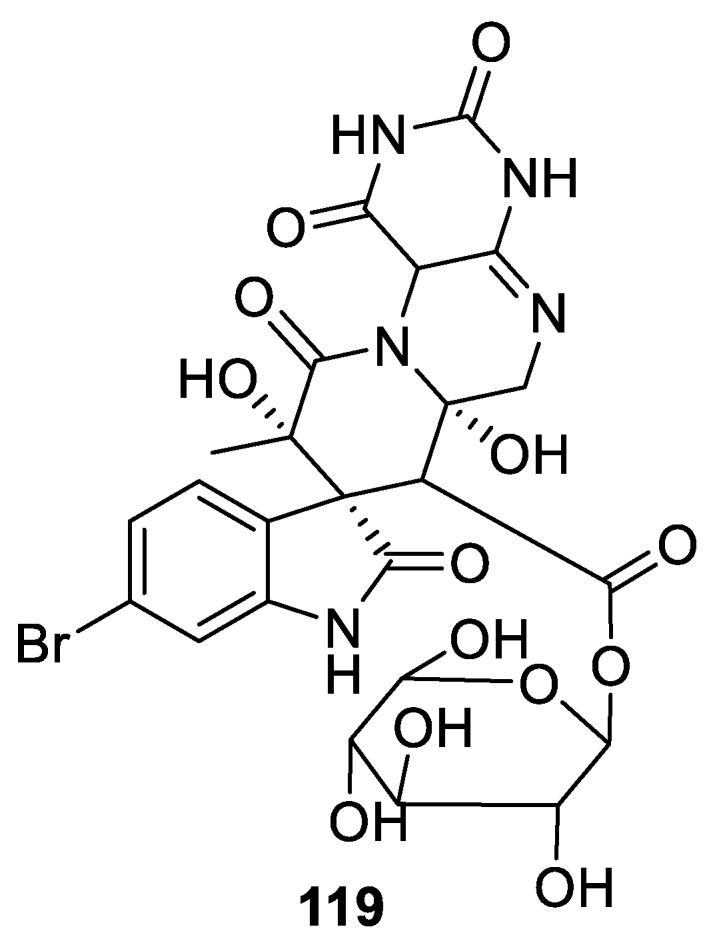
Surugatoxin isolated from the toxic Japanese ivory shell (*Babylonica japonica*).

**Figure 57 molecules-24-04165-f057:**
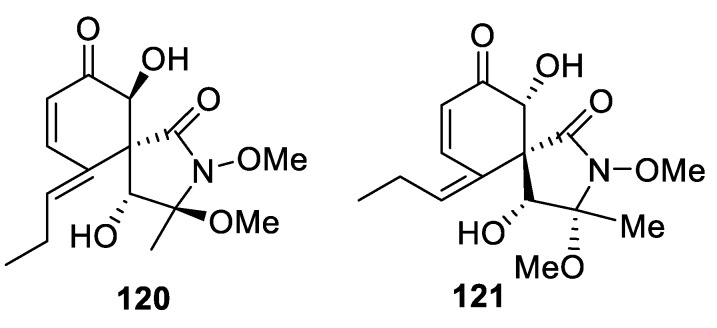
Structures of representative spirostaphylotrichins possessing a [4.5.0] spirocyclic motif.

**Figure 58 molecules-24-04165-f058:**
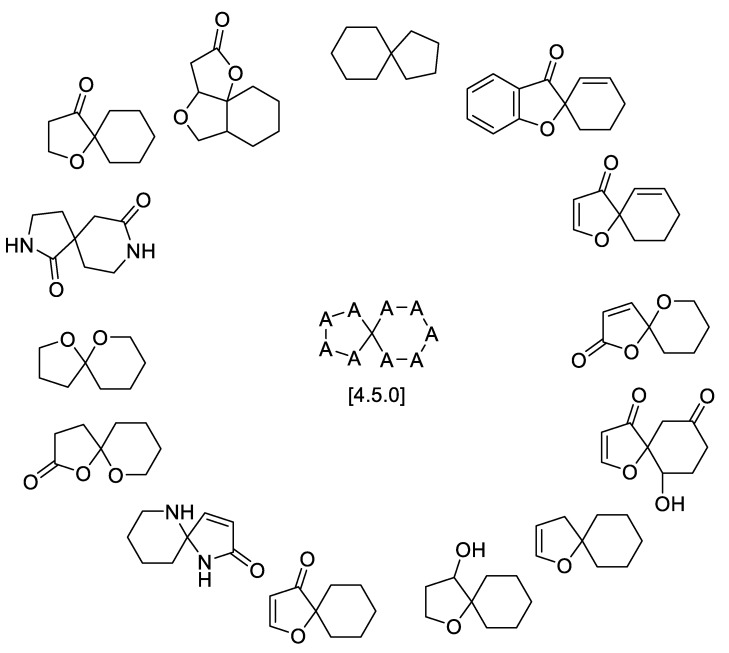
Current diversity of [4.5.0] spirocyclic scaffolds.

**Figure 59 molecules-24-04165-f059:**
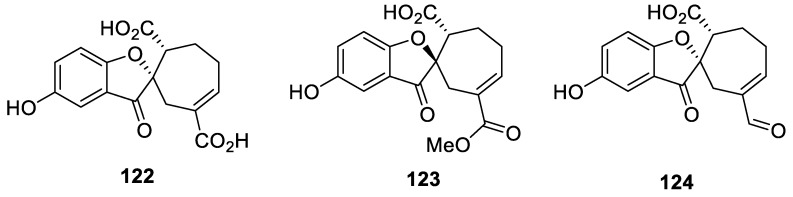
Spiro meroterpenoids spiroapplanatumines (**122**–**124**) isolated from fungus *Ganoderma applanatum*.

**Figure 60 molecules-24-04165-f060:**
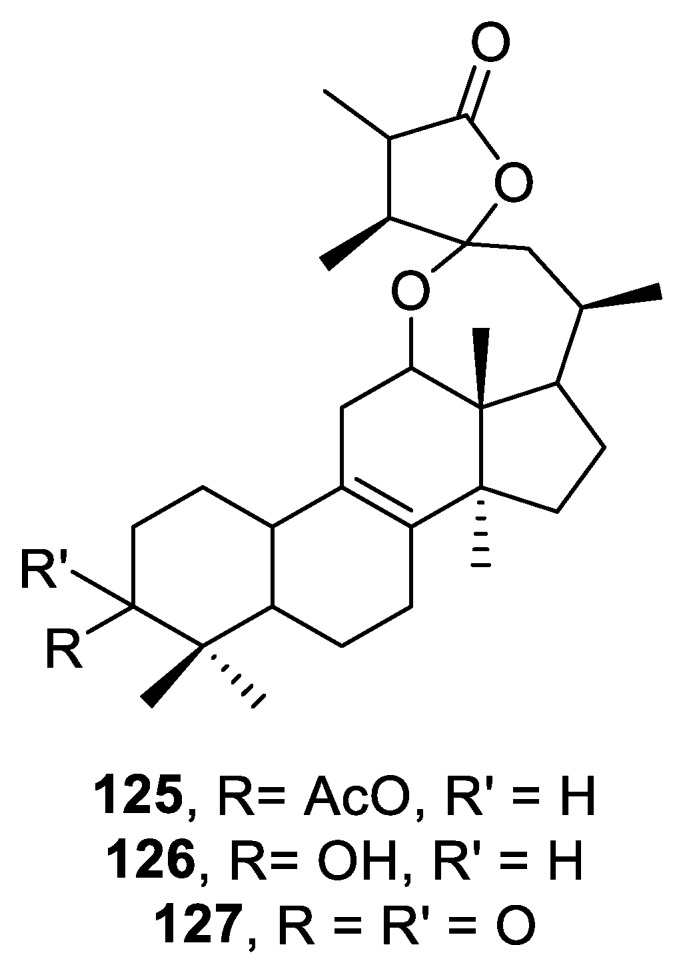
Fomlactones A–C possessing a [4.6.0] spirocyclic moiety.

**Figure 61 molecules-24-04165-f061:**
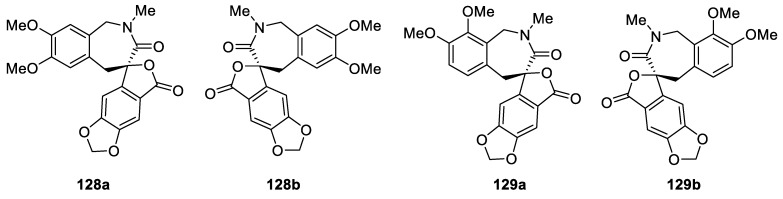
Enantiopure juglanaloid A (**128a**–**b**) and juglanaloid B (**129a**–**b**) isolated from *Juglans mandshurica* and further obtained by chiral separation.

**Figure 62 molecules-24-04165-f062:**
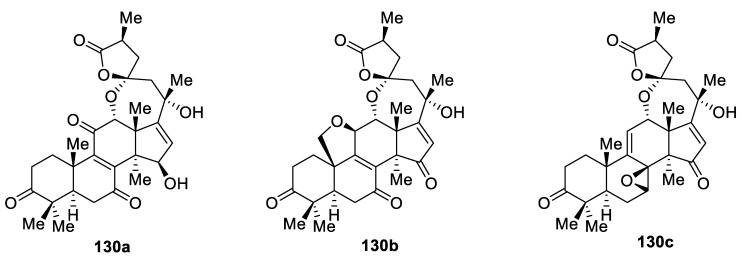
Lanostane-type triterpenoid spirolactones from *Ganoderma applanatum*.

**Figure 63 molecules-24-04165-f063:**
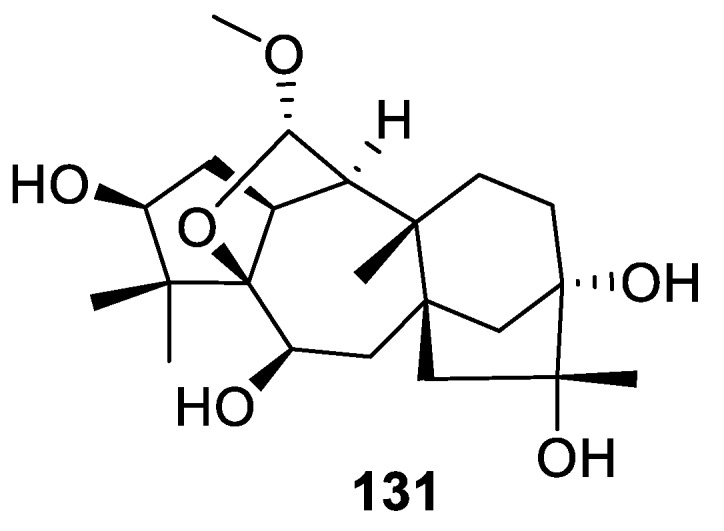
Structure of auriculatol A possessing a [4.6.0] spirocyclic motif.

**Figure 64 molecules-24-04165-f064:**
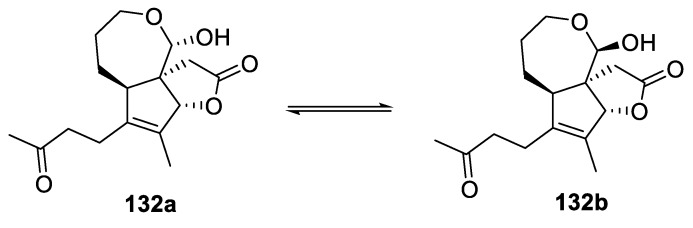
Structure of [4.6.0] spirocyclic seconoriridone A.

**Figure 65 molecules-24-04165-f065:**
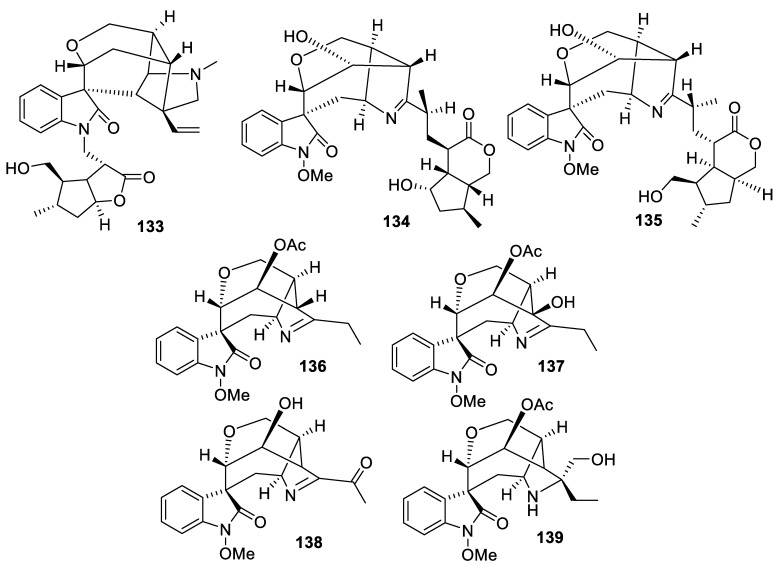
Structures of gelsenium alkaloids possessing a [4.6.0] spirocyclic system.

**Figure 66 molecules-24-04165-f066:**
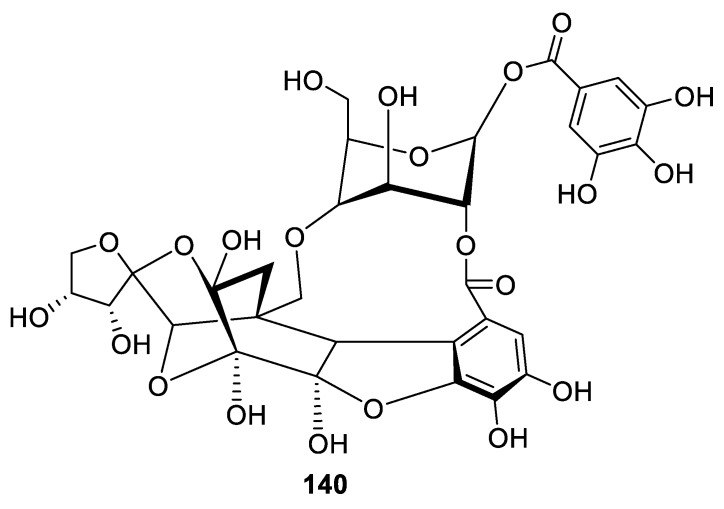
Natural product phyllanthunin possessing a [4.7.0] spirocyclic moiety isolated from *Phyllanthus emblica*.

**Figure 67 molecules-24-04165-f067:**
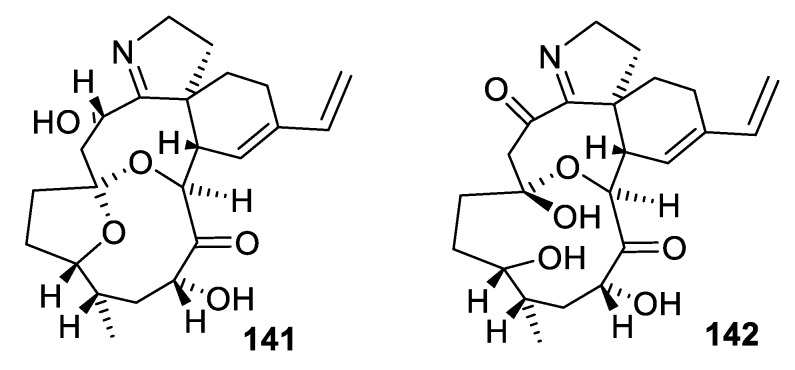
Portimines A and B isolated from *Vulcanodinium rugosum* containing both one [4.7.0] and one [4.5.0] spirocyclic motif.

**Figure 68 molecules-24-04165-f068:**
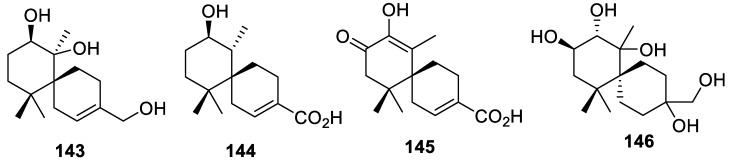
Spirocyclic chamigrane sesquiterpenes, merulinols B (**143**), C (**144**), E (**145**), and F (**146**).

**Figure 69 molecules-24-04165-f069:**
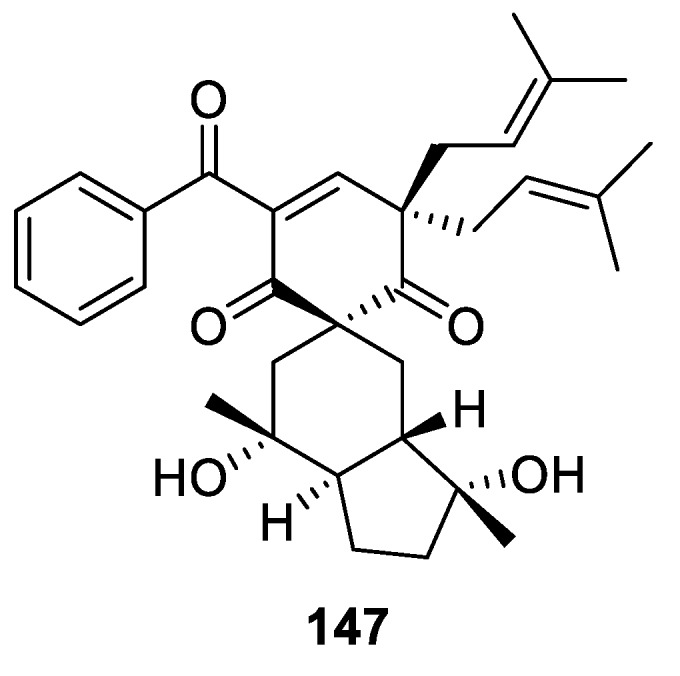
Hyperbeanol C isolated from *Hypericum beanie*.

**Figure 70 molecules-24-04165-f070:**
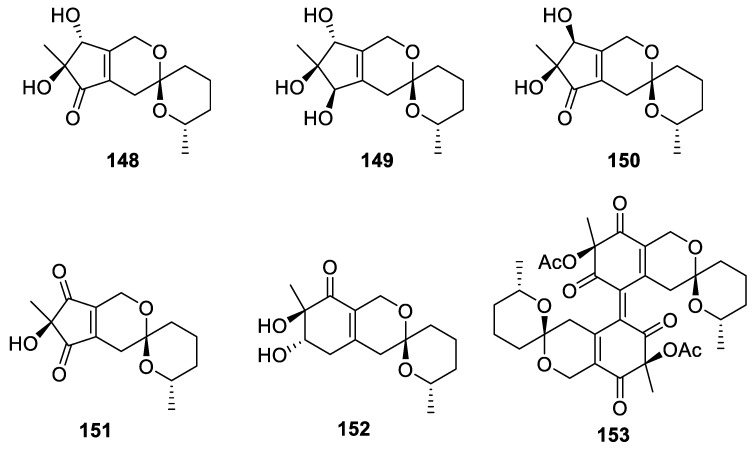
Thielavialides A−E (**148**–**152**) and pestafolide A (**153**).

**Figure 71 molecules-24-04165-f071:**
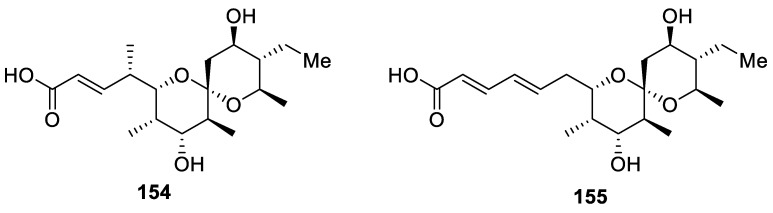
Pteridic acids C and F isolated from *Streptomyces* sp. SCSGAA 0027 possessing a 1,7-dioxaspiro[5.5.0]undecane motif.

**Figure 72 molecules-24-04165-f072:**
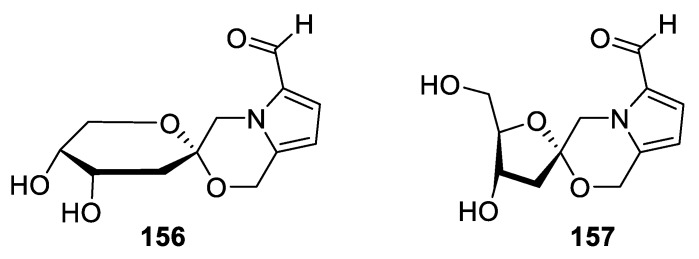
Pollenopyrroside A isolated from bee-collected *Brassica campestris* pollen.

**Figure 73 molecules-24-04165-f073:**
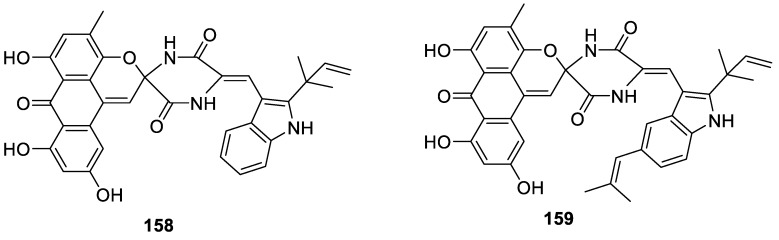
New spirocyclic piperazin-2,5-dione alkaloids isolated from *Aspergillus variecolor*.

**Figure 74 molecules-24-04165-f074:**
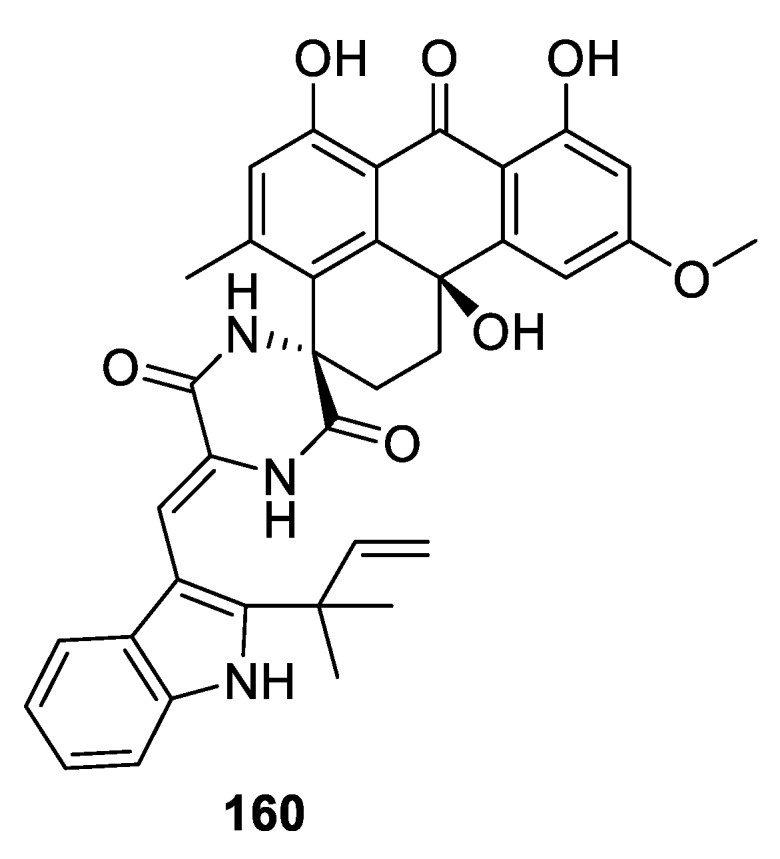
Spirocyclic piperazin-2,5-dione variecolortin B isolated from the marine-derived fungus *Eurotium* sp. SCSIO F452.

**Figure 75 molecules-24-04165-f075:**
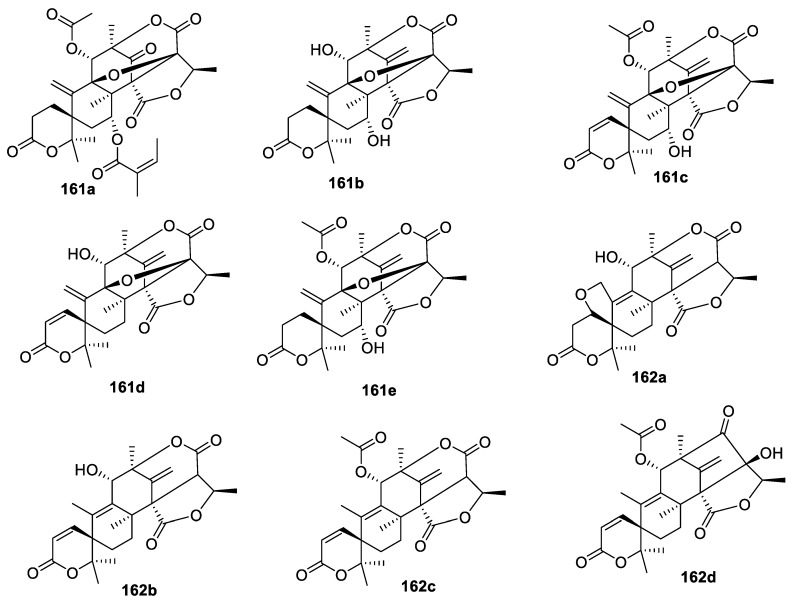
Bioactive [5.5.0] spirocyclic meroterpenoids isolated from mangrove-derived fungus *Penicillium* sp.

**Figure 76 molecules-24-04165-f076:**
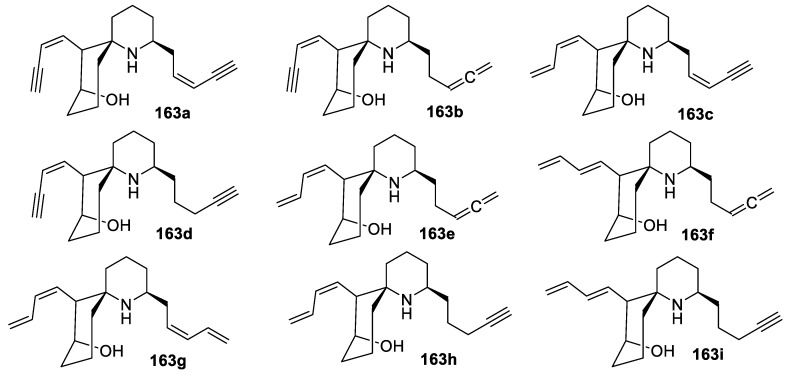
Alkaloids **163**–**i** of the histrionicotoxin class isolated from ant *Carebarella bicolor*.

**Figure 77 molecules-24-04165-f077:**
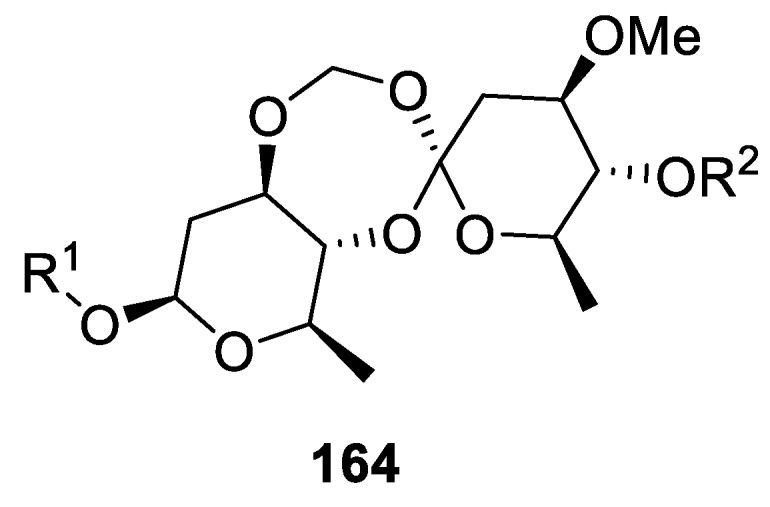
General structure of [5.6.0] spirocyclic orthoester periplosides.

**Figure 78 molecules-24-04165-f078:**
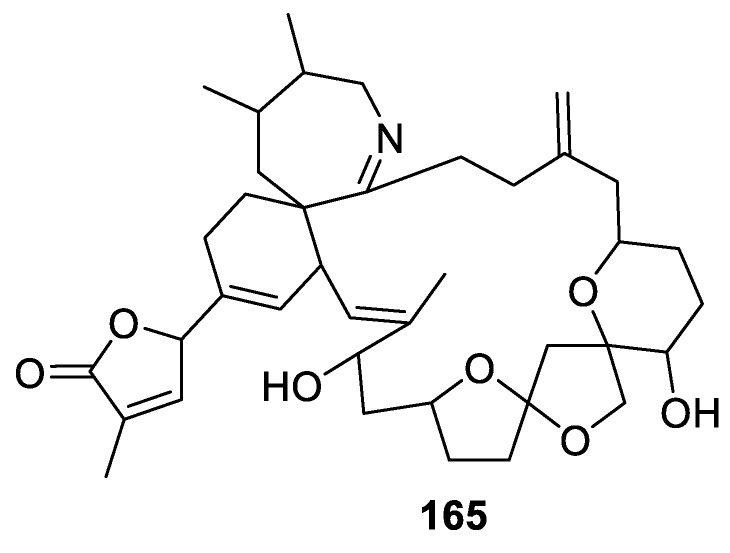
Spirolide G isolated from toxigenic dinoflagellate *Alexandrium ostenfeldii*.

**Figure 79 molecules-24-04165-f079:**
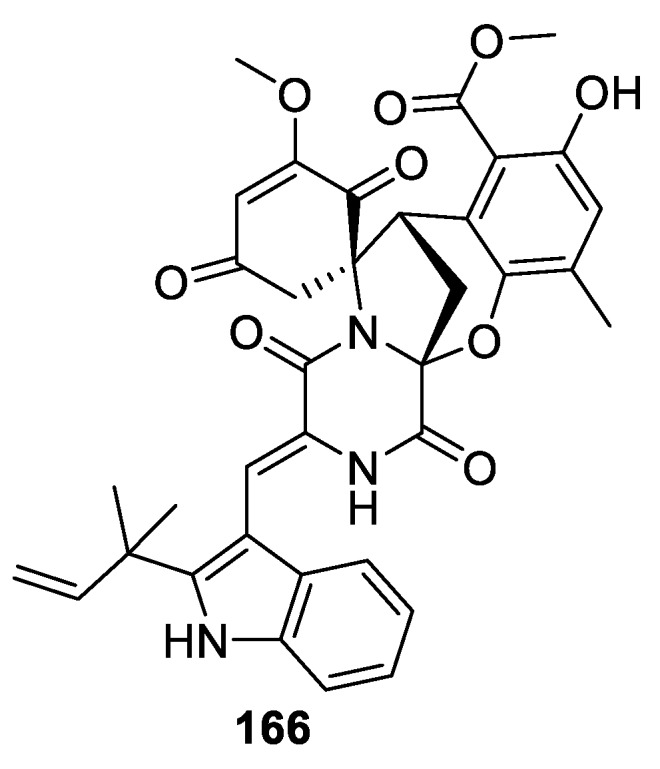
A [5.6.0] spirocyclic compound isolated from marine-derived fungus *Eurotium* sp. SCSIO F452.

**Figure 80 molecules-24-04165-f080:**
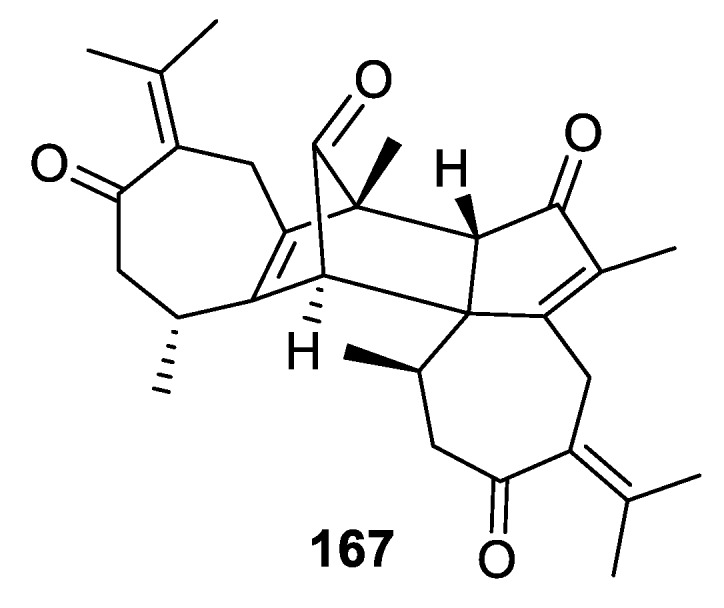
Vieloplain G isolated from *Xylopia vielana* containing a [5.6.0] spirocyclic scaffold.

**Figure 81 molecules-24-04165-f081:**
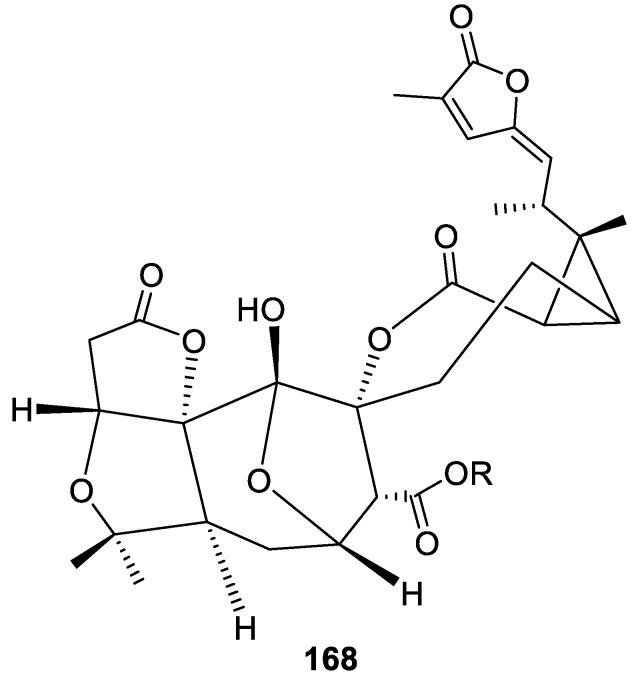
Spiroschincarin A isolated from the fruit of *Schisandra incarnate*.

**Figure 82 molecules-24-04165-f082:**
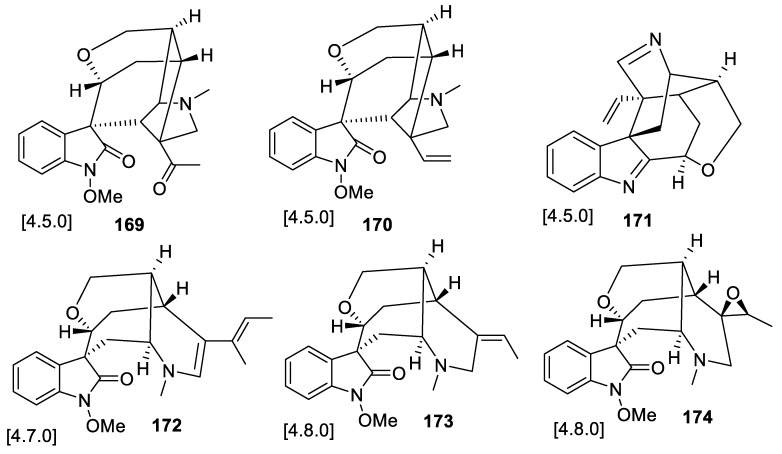
Structurally diverse spirocyclic frameworks isolated from a single plant species (*Gelsemium elegans*).

**Table 1 molecules-24-04165-t001:** Occurrence of various ring combinations in the spirocyclic natural products analyzed in this review.

Ring combinations	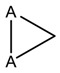	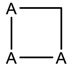	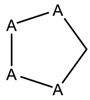	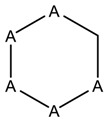	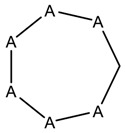	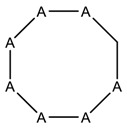
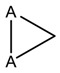	0	0	7	7	0	0
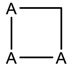	0	0	1	1	0	4
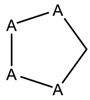	7	1	58	50	14	4
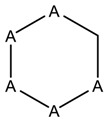	7	1	50	27	4	0
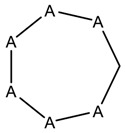	0	0	14	4	1	0
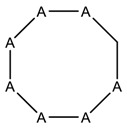	0	4	4	1	0	0

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
