# Peer review of "Spirocyclic Motifs in Natural Products"

_molecules, 2019, doi:10.3390/molecules24224165_

Round 1
Reviewer 1 Report
see attached file

Author Response
Thank you very much indeed for the valuable suggestion. We will include all suggested natural products after Figure 21, where it seems most appropriate to expand on spirooxiidoles.
Reviewer 2 Report
See attached

Author Response
Thank you very much indeed for the valuable suggestions. While we aimed at covering spirocyclic motifs occurring in the natural products as fully as possible (by condicting respective substructure searches in SciFinder), we appreciate that certain natural products were missed. We are, therefore, grateful for your help which will allow including more of the spriocyclic natural products (histrionicotoxins, duocarmycins, fredericamycin, spirostaphylotrichins, gelsemine) in the revised version of the review.
From our statistical analysis provided in Table 1 one can easily devise information on which spirocyclic motifs occur more frequently than others in natural products. Also, some of the spirocyclic systems have not been encountered at all. While it is true this information could be presented in alternative ways, we would like to leave it as is. This Table is already inspiring us (and will likely inspire others) as a guide to spirocyclic compound library design.
Other comments:
Compound numbers should be emboldened in the text, this was not always done. We have gone through the text with a fine comb during the revision to make sure all compound numbers are in bold. Thank you.
2. Couldn’t 19-22 be more accurately described as spiro[3.5.0] systems?
Yes, they certainly can. We have pointed this out in the revised version.
3. Structures 34-35 should be redrawn for clarity.
This will be done in the revised manuscript. Thank you.
4. Structure 121 should be redrawn for clarity. It seems that this is a [4.5.0] spirocycle?
This will be done in the revised manuscript. Thank you.
5. Structure 148 should be redrawn for clarity.
This will be done in the revised manuscript. Thank you.
A number of structures presented can be classified as spiroacetals, which distinguishes these compounds from other types of spirocycles. Might the authors want to consider grouping these compounds in a separate category?
We also considered this when planning the review. However, we decided to focus on structuring the review according to the size of the spirocyclic system [x.y.0] while singling out spiroacetals in a standalone group would require that several different [x.y.0] systems are included. Therefore, we would like to stay with the current structure.
Overall the manuscript was well-written, but there are several typos that should be corrected after careful proofreading.
Thank you. We have gone through the text with a fine comb during the revision to make sure all typos have been corrected.
Reviewer 3 Report
The manuscript molecules-631511 is devoted to the actual problem of medicinal chemistry. The reviewed article is interesting and theme of the article meets the scope of the journal. Work is performed at sufficient scientific level and has good quality. The authors have critically analyzed a large volume of scientific literature and successfully systematized it. The review is well organized. However, it needs minor revision before publication.
To improve the quality and perception of the manuscript I would suggest paying attention to following comments:
Abstract in general should be revised. The abstract should be more focused on the topic of the review. It should highlight the main points of the review. There are grammar and orthographical errors in the manuscript, which should be corrected.
After correction this manuscript may be accepted for publication. My decision is minor revision.
Author Response
Thank you for the very positive appraisal of our work. We duly noted the criticism regarding the phrasing of the Abstract and grammar and orthography flaws in the manuscript and have done our best to correct those throughout the text.
Round 2
Reviewer 2 Report
The manuscript is suitable for publication